# Synaptic density marker SV2A is reduced in schizophrenia patients and unaffected by antipsychotics in rats

Ellis Chika Onwordi[1,2,3,4], Els F. Halff [3], Thomas Whitehurst[1,3,4], Ayla Mansur[5], Marie-Caroline Cotel[6], Lisa Wells[7], Hannah Creeney[6], David Bonsall[7], Maria Rogdaki [1,2,3,4], Ekaterina Shatalina[1,2], Tiago Reis Marques[1,3,4], Eugenii A. Rabiner[7,8], Roger N. Gunn[5,7], Sridhar Natesan [1,3], Anthony C. Vernon [6,9] & Oliver D. Howes[1,2,3,4]*

Synaptic dysfunction is hypothesised to play a key role in schizophrenia pathogenesis, but this has not been tested directly in vivo. Here, we investigated synaptic vesicle glycoprotein 2A (SV2A) levels and their relationship to symptoms and structural brain measures using [11C]UCB-J positron emission tomography in 18 patients with schizophrenia and 18 controls. We found significant group and group-by-region interaction effects on volume of distribution ($V_T$). [11C]UCB-J $V_T$ was significantly lower in the frontal and anterior cingulate cortices in schizophrenia with large effect sizes (Cohen's $d = 0.8$-$0.9$), but there was no significant difference in the hippocampus. We also investigated the effects of antipsychotic drug administration on SV2A levels in Sprague-Dawley rats using western blotting, [3H]UCB-J autoradiography and immunostaining with confocal microscopy, finding no significant effects on any measure. These findings indicate that there are lower synaptic terminal protein levels in schizophrenia in vivo and that antipsychotic drug exposure is unlikely to account for them.

[1] Psychiatric Imaging Group, MRC London Institute of Medical Sciences, Hammersmith Hospital, London W12 0NN, UK. [2] Institute of Clinical Sciences (ICS), Faculty of Medicine, Imperial College London, London W12 0NN, UK. [3] Department of Psychosis Studies, Institute of Psychiatry, Psychology and Neuroscience, King's College London, De Crespigny Park, London SE5 8AF, UK. [4] South London and Maudsley NHS Foundation Trust, Camberwell, London SE5 8AF, UK. [5] Division of Brain Sciences, Imperial College London, The Commonwealth Building, Hammersmith Hospital, Du Cane Road, London W12 0NN, UK. [6] Department of Basic and Clinical Neuroscience, Institute of Psychiatry, Psychology and Neuroscience, Maurice Wohl Clinical Neuroscience Institute, King's College London, 5 Cutcombe Road, London SE5 9RT, UK. [7] Invicro Imaging Services, Burlington Danes Building, Du Cane Road, London W12 0NN, UK. [8] Centre for Neuroimaging Sciences, Institute of Psychiatry, Psychology and Neuroscience, King's College London, De Crespigny Park, London SE5 8AF, UK. [9] MRC Centre for Neurodevelopmental Disorders, King's College London, London SE1 1UL, UK. *email: oliver.howes@lms.mrc.ac.uk

Schizophrenia is a severe mental illness characterised by positive, negative and cognitive symptoms, and is a major cause of global disease burden[1,2]. Current treatments are inadequate for many patients[3,4], highlighting the need to understand the pathophysiology underlying the disorder to inform drug development[5,6].

It has been hypothesised that synaptic dysfunction is central to the pathophysiology of schizophrenia[7–9]. Genetic studies have found associations between schizophrenia and variants in genes encoding synaptic proteins[10,11]. A polymorphism in the synaptic vesicle glycoprotein 2A gene (SV2A) has been associated with increased schizophrenia risk[12], although this finding has not been replicated in genome-wide association studies. Associations between schizophrenia and variants in genes shown to mediate synaptic pruning in postnatal mouse neurodevelopment have been reported[13]. In addition, post-mortem studies have reported lower levels of dendritic spines[14,15] and of protein and mRNA levels of a number of synaptic markers, including synaptic vesicle proteins such as synaptophysin and synaptobrevin[16–19], and postsynaptic markers such as PSD-95[17,20], in schizophrenia. Synaptophysin is enriched in presynaptic nerve terminals[21] where, along with synaptobrevin and SV2A, it is involved in regulating vesicle exocytosis into synapses[22,23]. Recent meta-analyses of post-mortem studies showed significantly lower synaptophysin levels in the frontal and cingulate cortices and hippocampus with moderate-to-large effect size[8] and lower postsynaptic element levels in schizophrenia relative to controls[24]. Furthermore, human-derived neural cultures have shown elevated, complement-dependent elimination of synaptic structures[25], lower neurite number and neuronal connectivity[26] and synaptic vesicle release deficits[27] in schizophrenia relative to healthy controls. Further, albeit indirect, evidence for altered synaptic function comes from in vivo neuroimaging studies which show lower brain volumes[28] and altered functional connectivity in schizophrenia relative to controls[29], which may reflect altered synaptic density and/or function in schizophrenia (although there are other potential explanations for these findings). Thus, there are converging lines of evidence implicating synaptic dysfunction in the pathophysiology of schizophrenia, and specifically for lower levels of synaptic proteins.

However, the post-mortem findings may be affected by a number of potential confounds, including differences in post-mortem interval, cause of death and antipsychotic medication exposure[30]. Thus, it is unknown whether synaptic protein levels are altered in vivo. The recent development of a positron emission tomography (PET) radiotracer specific for SV2A now permits in vivo investigation of presynaptic protein levels[31]. Therefore, we conducted parallel clinical and preclinical studies to address several key questions. In our clinical study, we used [11C] UCB-J PET to test the synaptic hypothesis of schizophrenia, anticipating that its volume of distribution ($V_T$) would be lower in patients with schizophrenia relative to controls in the frontal cortex (FC), anterior cingulate cortex (ACC) and hippocampus, regions where synaptic protein levels are lower post-mortem in schizophrenia on meta-analysis[8], and that levels would be inversely associated with symptom severity.

Furthermore, we performed an exploratory analysis to test whether effects are generalised to other brain regions (occipital, parietal and temporal lobes, dorsolateral prefrontal cortex, thalamus and amygdala) where structural and/or functional alterations have been identified in schizophrenia using in vivo neuroimaging[28,32,33], and we calculated distribution volume ratio (DVR) in each region of interest (ROI), using the centrum semiovale as a pseudoreference region for estimates of non-displaceable binding of [11C]UCB-J[31]. In our preclinical study, we used a well-validated rat model of clinically comparable antipsychotic drug exposure to investigate whether chronic antipsychotic drug exposure at clinically relevant doses alters SV2A protein levels and/or specific binding in drug-naïve rats, predicting there would be no effect given the evidence that chronic haloperidol (HAL) exposure does not impact on synaptophysin levels in the rat FC[34,35]. We report here the first direct in vivo evidence for synaptic dysfunction in schizophrenia together with results from preclinical experiments suggesting that this dysfunction is not explained by antipsychotic drug exposure at clinically relevant doses.

## Results

Thirty-six volunteers ($n = 18$ with schizophrenia [SCZ, 15 male and 3 female] and 18 healthy volunteers [HV, 15 male and 3 female]) completed the clinical study. The groups were not significantly different in age (mean [standard error of the mean, SEM] years of SCZ group = 41.5 [2.7]; HV group = 38.7 [3.1]; Kolmogorov–Smirnov $D = 0.28$, $p = 0.49$), activity injected (mean [SEM] MBq in SCZ = 237.4 [11.76]; HV = 257.1 [6.49]; Kolmogorov–Smirnov $D = 0.28$, $p = 0.49$) or [11C]UCB-J plasma-free fraction (mean [SEM] $f_P$ in SCZ = 0.24 [0.005]; HV = 0.25 [0.006]; $t = 0.63$, df = 34, $p = 0.53$).

All volunteers with schizophrenia were on antipsychotic medication (mean [SEM] chlorpromazine-equivalent dose = 443.5 [89.6] mg/day, Supplementary Table 1). None of the volunteers with schizophrenia had co-morbid DSM-5 psychiatric diagnoses.

**[11C]UCB-J $V_T$ across groups in a priori ROIs.** There was a significant effect of group (two-way ANOVA: $F_{1,34} = 6.170$, $p = 0.02$) and ROI ($F_{2,68} = 426.0$, $p < 0.0001$) on [11C]UCB-J $V_T$. Furthermore, there was a significant group-by-ROI interaction effect on [11C]UCB-J $V_T$ ($F_{2,68} = 7.472$, $p = 0.001$). Post-hoc analyses with false-discovery rate (FDR) adjustment revealed that mean [SEM] [11C]UCB-J $V_T$ (ml/cm$^3$) was significantly reduced in the SCZ relative to the HV group in the frontal cortex, FC (SCZ = 16.93 [0.80]; HV = 19.50 [0.64]; $t = 2.51$, df = 34.0, $p = 0.03$), and in the anterior cingulate cortex, ACC (SCZ = 19.55 [0.75]; HV = 22.49 [0.72]; $t = 2.83$, df = 34.0, $p = 0.02$), with large effect sizes (Cohen's $d = 0.8$ and 0.9, respectively), as shown in Fig. 1. However, there was no significant difference between groups in [11C]UCB-J $V_T$ in the hippocampus (mean [SEM] $V_T$ SCZ = 14.09 [0.59]; HV = 15.44 [0.50]; $t = 1.75$, df = 34.0, $p = 0.09$, Cohen's $d = 0.6$). Figure 2 shows mean parametric [11C] UCB-J $V_T$ brain images for the SCZ and HV groups.

**Relationship between [11C]UCB-J $V_T$ and grey matter volume.** There was a significant effect of ROI (two-way ANOVA: $F_{2,68} = 1835$, $p < 0.0001$) but not of group ($F_{1,34} = 2.10$, $p = 0.16$) or group-by-ROI interaction effect ($F_{2,68} = 1.85$, $p = 0.16$), on corrected grey matter volume (GMV) (Supplementary Fig. 1). There were no significant associations between [11C]UCB-J $V_T$ and corrected GMV in any of the ROIs, neither in the combined sample nor in HV nor SCZ groups analysed separately (Table 1).

**Relationship between [11C]UCB-J $V_T$ and clinical variables.** There were no significant relationships between [11C]UCB-J $V_T$ in the FC, ACC or hippocampus and chlorpromazine-equivalent dose, Positive and Negative Syndrome Scale (PANSS) total, PANSS-positive, PANSS-negative or PANSS general scores or duration of illness (Table 2). We conducted exploratory analyses to explore the potential influence of smoking, concomitant medication and clozapine treatment on $V_T$. These showed no significant effect of these variables on $V_T$ (see Supplementary Notes 1, 2 and 3).

**Exploratory analysis of [11C]UCB-J $V_T$ in other regions**. There was a significant effect of group (two-way ANOVA: $F_{1,34} = 6.362$, $p = 0.02$) and ROI ($F_{5,170} = 125.2$, $p < 0.0001$) on [11C]UCB-J $V_T$. There was no significant group-by-ROI interaction effect on [11C]UCB-J $V_T$ ($F_{5,170} = 1.148$, $p = 0.34$). Post-hoc analyses with FDR

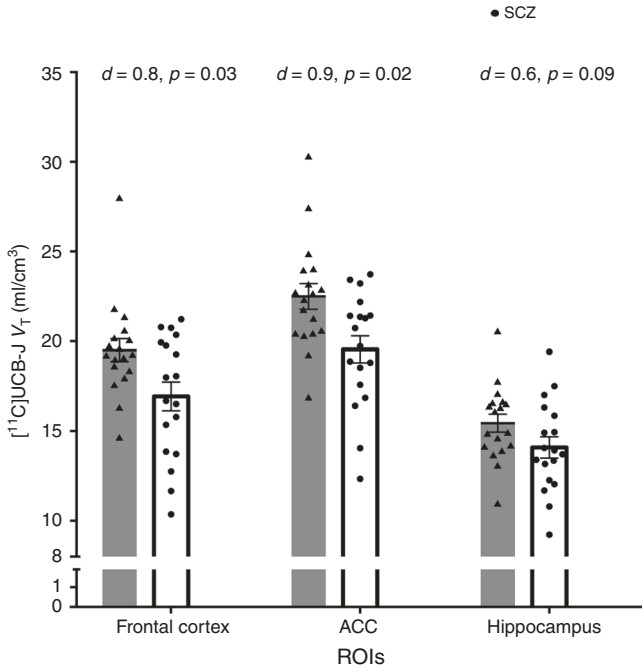

**Fig. 1 Regional mean [11C]UCB-J distribution volume ($V_T$) by group.** Grey bars depict regional mean $V_T$ in the healthy volunteer (HV) group, and triangles represent individual HV $V_T$ ($n = 18$). Hollow bars depict mean $V_T$ in the schizophrenia (SCZ) group, and circles indicate individual SCZ patient $V_T$ ($n = 18$). $p$ Values reported here are FDR-adjusted $p$ values. [11C]UCB-J $V_T$ was significantly reduced with large effect sizes (Cohen's $d > 0.8$) in the SCZ compared to the HV group in the frontal cortex and anterior cingulate cortex (ACC). [11C]UCB-J $V_T$ was not significantly altered in the hippocampus. Error bars indicate standard error of the mean.

adjustment revealed that mean [SEM] [11C]UCB-J $V_T$ (ml/cm³) was significantly reduced in the SCZ relative to the HV group in the dorsolateral prefrontal cortex (SCZ = 17.20 [0.80]; HV = 19.83 [0.59]; $t = 2.63$, df = 34.0, $p = 0.03$, Cohen's $d = 0.9$); temporal lobe (SCZ = 18.05 [0.68]; HV = 20.64 [0.68]; $t = 2.69$, df = 34.0, $p = 0.03$, Cohen's $d = 0.9$); occipital lobe (SCZ = 16.98 [0.64]; HV = 19.25 [0.73]; $t = 2.34$, df = 34.0, $p = 0.04$, Cohen's $d = 0.8$); parietal lobe (SCZ = 17.25 [0.70]; HV = 19.42 [0.75]; $t = 2.12$, df = 34.0, $p = 0.04$, Cohen's $d = 0.7$), thalamus (SCZ = 13.12 [0.54]; HV = 15.17 [0.62]; $t = 2.51$, df = 34.0, $p = 0.03$, Cohen's $d = 0.8$) and amygdala (SCZ = 17.09 [0.64]; HV = 18.78 [0.64]; $t = 2.09$, df = 34.0, $p = 0.03$, Cohen's $d = 0.7$; Supplementary Fig. 2).

**[11C]UCB-J $V_T$ across groups in centrum semiovale**. The groups were not significantly different in [11C]UCB-J $V_T$ (ml/cm³) in the centrum semiovale (mean [SEM] $V_T$ (ml/cm³) in SCZ group = 6.06 [0.34], HV group = 5.66 [0.14], $t = 1.10$, df = 34, $p = 0.28$ [two-tailed independent samples $t$-test]).

**[11C]UCB-J DVR across groups in a priori ROIs**. Figure 3 shows the DVRs by group. There was a significant effect of group (two-way ANOVA: $F_{1,34} = 8.1$, $p = 0.007$) and of ROI ($F_{2,68} = 510.9$, $p < 0.0001$) on [11C]UCB-J DVR. Furthermore, there was a significant group-by-ROI interaction effect on [11C]UCB-J DVR ($F_{2,68} = 7.97$, $p = 0.0008$). Post-hoc analyses with FDR adjustment revealed that mean (SEM) [11C]UCB-J DVR was significantly lower in the FC (SCZ = 2.93 [0.17]; HV = 3.48 [0.09]; $t = 2.89$, df = 34.0, $p = 0.01$, Cohen's $d = 1.0$), ACC (SCZ = 3.39 [0.17]; HV = 3.99 [0.09]; $t = 3.05$, df = 34.0, $p = 0.01$, Cohen's $d = 1.0$) and the hippocampus (SCZ = 2.40 [0.12]; HV = 2.74 [0.07]; $t = 2.32$, df = 34.0, $p = 0.03$, Cohen's $d = 0.8$; Fig. 3).

**[11C]UCB-J DVR across groups in exploratory ROIs**. There was a significant effect of group (two-way ANOVA: $F_{1,34} = 8.47$, $p = 0.0063$) and of ROI ($F_{5,170} = 153.4$, $p < 0.0001$) on [11C]UCB-J DVR. There was no significant group-by-ROI interaction effect on [11C]UCB-J DVR ($F_{5,170} = 1.75$, $p = 0.13$). Post-hoc analyses with FDR adjustment revealed that mean [SEM] [11C]UCB-J DVR was significantly lower in the occipital lobe (SCZ = 2.93 [0.15]; HV = 3.43 [0.10]; $t = 2.72$, df = 34.0, $p = 0.02$, Cohen's

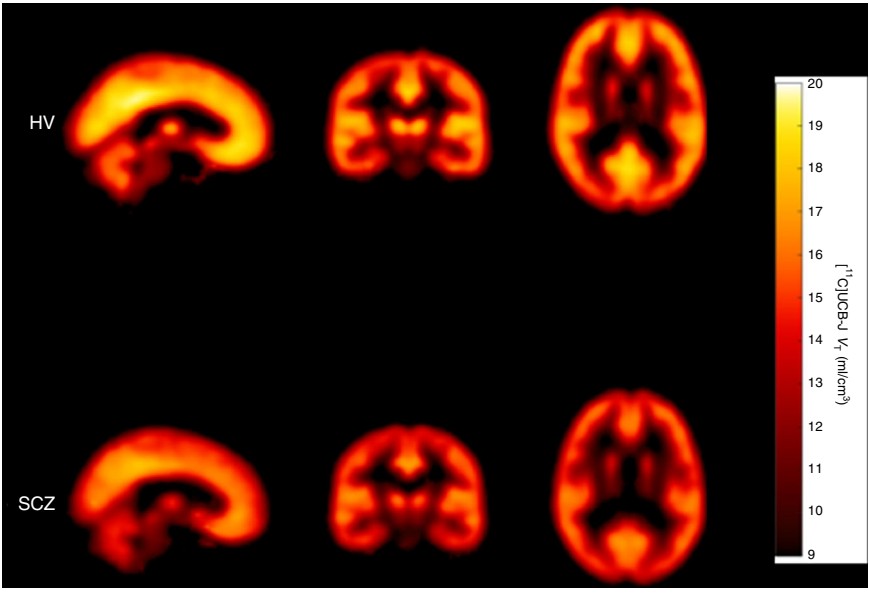

**Fig. 2 Mean parametric [11C]UCB-J $V_T$ images from healthy volunteer (HV) and schizophrenia (SCZ) groups.** Colour bar indicates [11C]UCB-J $V_T$.

**Table 1 Associations between corrected grey matter volume and [$^{11}$C]UCB-J $V_T$.**

| | Full sample ($n = 36$) correlation coefficient | $p$ value | HV ($n = 18$) correlation coefficient | $p$ value | SCZ ($n = 18$) correlation coefficient | $p$ value |
|---|---|---|---|---|---|---|
| Hippocampus | *0.21* | 0.21 | *0.06* | 0.82 | *0.26* | 0.30 |
| Frontal cortex | *0.20* | 0.25 | *0.01* | 0.95 | 0.13 | 0.61 |
| Anterior cingulate cortex | *0.19* | 0.26 | −0.005 | 0.99 | *0.32* | 0.20 |

Relationships were explored in the hippocampus, frontal cortex and anterior cingulate cortex, in the full sample and in the separated schizophrenia (SCZ) and healthy volunteer (HV) groups. All correlation coefficients reported are Pearson product-moment correlation coefficients, except where use of italics signifies Spearman's rank correlation coefficients for nonparametric data. There were no significant associations between grey matter volume and [$^{11}$C]UCB-J $V_T$ in any of the regions of interest

**Table 2 Associations between clinical variables and [$^{11}$C]UCB-J $V_T$.**

| | PANSS total correlation coefficient | $p$ Value | PANSS-positive correlation coefficient | $p$ Value | PANSS-negative correlation coefficient | $p$ Value | PANSS general correlation coefficient | $p$ Value | Chlorpromazine-equivalent dose correlation coefficient | $p$ Value | Duration of illness correlation coefficient | $p$ Value |
|---|---|---|---|---|---|---|---|---|---|---|---|---|
| Hippocampus | −0.09 | 0.71 | −0.11 | 0.69 | −0.10 | 0.72 | −0.04 | 0.89 | *−0.36* | 0.14 | *−0.15* | 0.56 |
| Frontal cortex | −0.13 | 0.61 | −0.07 | 0.80 | −0.09 | 0.73 | −0.14 | 0.60 | *−0.34* | 0.17 | *−0.29* | 0.24 |
| Anterior cingulate cortex | −0.04 | 0.87 | 0.02 | 0.93 | −0.09 | 0.73 | −0.04 | 0.89 | *−0.32* | 0.20 | *−0.33* | 0.17 |

Associations between PANSS scores (total and positive, negative and general subscale scores), chlorpromazine-equivalent dose and duration of illness and [$^{11}$C]UCB-J $V_T$ were explored in the schizophrenia group ($n = 18$) in the hippocampus, frontal cortex and anterior cingulate cortex. There were no significant associations between PANSS scores, chlorpromazine-equivalent dose or duration of illness and [$^{11}$C]UCB-J $V_T$ in any of the regions of interest. All correlation coefficients reported are Pearson product–moment correlation coefficients, except where use of italics signifies Spearman's rank correlation coefficients for nonparametric data

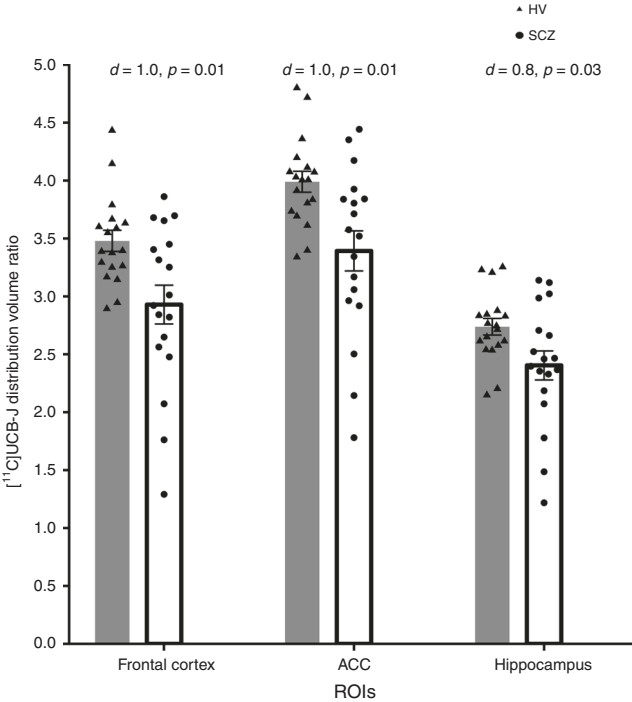

**Fig. 3 Regional mean [$^{11}$C]UCB-J distribution volume ratio (DVR) group.** Grey bars depict regional mean DVR in the healthy volunteer (HV) group, and triangles represent individual HV DVR ($n = 18$). Hollow bars depict mean DVR in the schizophrenia (SCZ) group, and circles indicate individual SCZ patient DVR ($n = 18$). FDR-adjusted $p$ values from post-hoc $t$-tests are reported here. [$^{11}$C]UCB-J DVR was significantly reduced with large effect sizes (Cohen's $d > 0.8$) in the SCZ compared to the HV group in the frontal cortex, anterior cingulate cortex (ACC) and the hippocampus. Error bars indicate standard error of the mean.

$d = 0.9$); parietal lobe (SCZ = 2.97 [0.16]; HV = 3.46 [0.11]; $t = 2.61$, df = 34.0, $p = 0.02$, Cohen's $d = 0.9$); temporal lobe (SCZ = 3.10 [0.15]; HV = 3.69 [0.09]; $t = 3.36$, df = 34.0, $p = 0.01$, Cohen's $d = 1.1$); dorsolateral prefrontal cortex (SCZ = 2.99

**Table 3 Plasma levels of haloperidol (HAL) and olanzapine (OLZ) in treated rats.**

| Cohort | 0.5 mg/kg/day HAL | 2 mg/kg/day HAL | 7.5 mg/kg/day OLZ |
|---|---|---|---|
| 1 | 2.96 ± 0.52 ng/ml | 12.2 ± 1.96 ng/ml | n.a. |
| 2 | 3.03 ± 0.94 ng/ml | n.a. | n.a. |
| 3. | n.a. | n.a. | 15.5 ± 5.54 ng/ml |

"n.a." means "not applicable"

[0.17]; HV = 3.53 [0.09]; $t = 2.89$, df = 34.0, $p = 0.02$, Cohen's $d = 1.0$); thalamus (SCZ = 2.28 [0.13]; HV = 2.74 [0.10]; $t = 2.77$, df = 34.0, $p = 0.02$, Cohen's $d = 0.9$); and amygdala (SCZ = 2.95 [0.14]; HV = 3.35 [0.08]; $t = 2.54$, df = 34.0, $p = 0.02$, Cohen's $d = 0.8$, Supplementary Fig. 3).

**Chronic antipsychotic drug exposure in rats.** Administration of the antipsychotic drugs haloperidol (HAL) and olanzapine (OLZ) by osmotic minipumps for 28 days achieved clinically relevant plasma levels in rats, comparable with previous studies (Table 3)[36–38].

**SV2A protein levels in the rat frontal cortex.** We first examined total SV2A protein levels in synaptosomes isolated from rat frontal cortex (FC) samples (comprising both the prefrontal and anterior cingulate cortices [PFC and ACC]) using western blotting. This was based on the finding that regional protein levels of SV2A are highly correlated with in vivo specific binding of [$^{11}$C] UCB-J[31] and the main finding of our clinical study that [$^{11}$C] UCB-J binding is significantly reduced in the frontal and anterior cingulate cortices of volunteers with schizophrenia who were receiving antipsychotic medication, as compared to controls. A protein band of ~100 kDa corresponding to SV2A was observed using synaptosomes prepared from rat FC (Fig. 4a). The primary antibody used has been validated previously both in SV2A knockout mice and with blocking peptides[39]. Chronic exposure for 28 days to two different doses of HAL did not significantly affect the protein levels of SV2A (relative to GAPDH) in the rat FC as

compared to vehicle controls (Kruskal–Wallis: $p = 0.71$); Fig. 4b. Furthermore, analyses of uncorrected SV2A protein levels in the rat FC showed no significant effect of HAL (Supplementary Note 4). See Supplementary Fig. 4 for unprocessed western blot images of SV2A and GAPDH protein levels in synaptosome fractions purified from the rat FC.

**[³H]UCB-J specific binding in the rat FC.** Using brain tissue from the opposite hemisphere of a subset of the animals ($n = 15$) in which western blots were carried out, we next measured the specific binding of [³H]UCB-J using ex vivo autoradiography in the PFC and ACC. In agreement with prior in vivo studies[31]

using [¹¹C]UCB-J, the specific binding of [³H]UCB-J is clearly higher in grey as compared to white matter (Fig. 4c). There were, however, no statistically significant effects of HAL administration on [³H]UCB-J specific binding (two-way ANOVA: $F_{2,12} = 1.451$, $p = 0.27$); Fig. 4d.

**Cellular localisation of SV2A in the rat FC.** To confirm the aforementioned results and evaluate the cellular localisation of SV2A, fluorescence immunohistochemistry was carried out on fixed brain tissue from a second cohort of rats chronically exposed to either vehicle or HAL (0.5 mg/kg/day) for 28 days. Consistent with the ex vivo autoradiography and prior in vivo work[31], SV2A

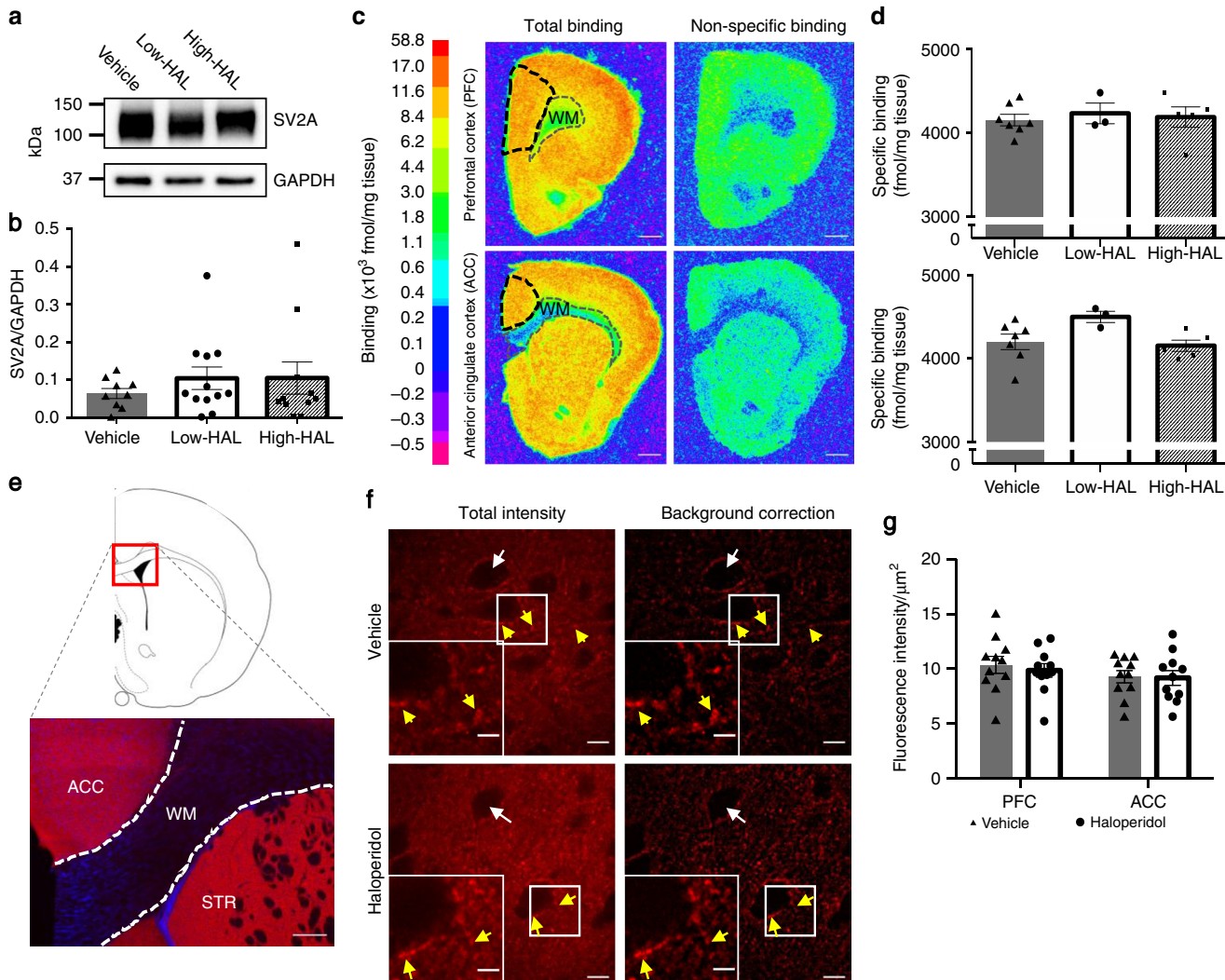

**Fig. 4 Chronic haloperidol administration does not affect SV2A levels in the rat frontal cortex (FC). a** Example of western blots showing total SV2A levels in synaptosomes isolated from the FC. **b** Quantification of total SV2A levels on western blot, normalised to GAPDH, showed no significant changes upon chronic haloperidol treatment (Kruskal–Wallis test, $p = 0.71$). **c** Autoradiography images of brain sections incubated with 12.5 nM [³H]UCB-J in absence (left) or presence (right) of levetiracetam (a drug that binds to SV2A). Black boxes indicate which ROI was selected to measure tracer binding in the prefrontal cortex, PFC (top), or anterior cingulate cortex, ACC (bottom). WM, white matter. Colour bar indicates [³H]UCB-J binding levels. Scale bar, 1 mm **d** Quantification of [³H]UCB-J binding in PFC (top) and ACC (bottom), indicating no significant changes upon chronic haloperidol treatment (two-way ANOVA, $F_{2,12} = 1.451$, $p = 0.27$). **e** Top: schematic of brain section at Bregma 1.1 mm. Bottom: low magnification fluorescent image of a rat brain section immunostained for SV2A (red) and DAPI (blue). Scale bar, 250 μm. ACC, anterior cingulate cortex; WM, white matter; STR, striatum. **f** High-magnification confocal images of brain sections from vehicle- (top) or haloperidol- (bottom) treated animals, showing punctate immunostaining for SV2A. Both raw images (left hand side) as well as background corrected images (right hand side) are shown. White arrows indicate a lack of SV2A punctae in the cell body; yellow arrows indicate examples of presynaptic terminals at perisomatic synapses and proximal dendrites. Scale bar, 10 μm (large field of view) or 4 μm (zoomed insets). **g** Quantification of background corrected SV2A intensity values. There were no significant differences in SV2A immunostaining intensity between vehicle- and haloperidol-treated groups (two-way ANOVA with Bonferroni's multiple comparison correction, $F_{1,20} = 0.1412$, $p = 0.71$). Error bars indicate standard error of the mean.

staining was high in grey matter, but negligible in white matter (Fig. 4e). SV2A staining was also absent in neuronal cell bodies (white arrows in Fig. 4f) but high at the perisomatic region (including the border of a proximal dendrite, yellow arrows in Fig. 4f). Total SV2A intensity after background correction varied significantly by region (two-way ANOVA: $F_{1,20} = 5.112$ $p = 0.035$), but did so comparably between groups with no significant effect of treatment ($F_{1,20} = 0.1412$, $p = 0.71$) or treatment-by-region interaction ($F_{1,20} = 0.2513$, $p = 0.62$); Fig. 4g.

Finally, to test whether the findings with HAL can be generalised to other antipsychotic drugs, we explored the effects of chronic OLZ administration on SV2A immunostaining intensity in the rat PFC and ACC (Supplementary Fig. 5). In agreement with the results from HAL treatment, chronic administration of OLZ also did not induce any significant differences in SV2A immunostaining intensity (two-way ANOVA; effect of treatment: $F_{1,14} = 0.0018$, $p = 0.97$; region: $F_{1,14} = 2.01$, $p = 0.18$; treatment-by-region interaction: $F_{1,14} = 1.87$, $p = 0.19$).

## Discussion

Our main finding is that [11C]UCB-J volume of distribution ($V_T$), an in vivo marker of synaptic vesicle glycoprotein 2A levels and proxy marker of synaptic nerve terminal density, is significantly lower in schizophrenia relative to controls in the FC and ACC with large effect sizes (Cohen's $d = 0.8–0.9$). Furthermore, we did not detect a significant effect of a 28-day antipsychotic drug exposure at clinically comparable doses on either SV2A protein levels or [3H]UCB-J specific binding in the PFC or ACC of the naïve rat brain. Our results confirm and extend post-mortem evidence for lower levels of presynaptic protein markers in schizophrenia[8] by demonstrating lower levels of SV2A in the FC and ACC in vivo in schizophrenia for the first time.

Moreover, in contrast to post-mortem studies, we were able to examine multiple regions in each individual, and find evidence for regional differences in the magnitude of effects, given the finding of a significant group-by-ROI interaction effect. We found lower [11C]UCB-J $V_T$ across multiple brain regions, but also evidence that frontal cortical regions may be more affected than hippocampus. The results were largely the same when using an alternative outcome measure, the DVR, which uses a brain region, the centrum semiovale, that is thought to be largely devoid of SV2A as a reference region to adjust for non-specific binding[31]. The one exception was the hippocampus, where DVR was significantly lower in the schizophrenia group relative to controls, albeit the effect size in the hippocampus was not as large as those in the other regions. This difference between the $V_T$ and DVR results for the hippocampus may reflect the fact that DVR values show lower variability and so are likely to have greater sensitivity to detect group differences[40], and our study lacked sufficient power to detect smaller effect size differences in hippocampal $V_T$. Thus, our finding that effects are less marked in the hippocampus relative to the FC warrants confirmation in future studies.

The exploratory analysis of additional brain regions showed significantly lower SV2A levels in the dorsolateral prefrontal and temporal cortices and occipital lobe in schizophrenia, consistent with other evidence implicating synaptic pathology in these regions in schizophrenia[14,15,19,41,42], but not consistent with meta-analytic post-mortem findings for synaptophysin in these regions[8]. The discrepancies between our findings and the post-mortem findings in these regions may reflect the fact we measured SV2A binding whilst the post-mortem studies measured other synaptic proteins, or could reflect the fact that our sample was younger and had a shorter illness duration than most of the patients in the post-mortem analyses, and/or the influence of post-mortem interval on

synaptic protein levels[8]. Our null findings in our preclinical study are in agreement with, and extend, previous preclinical findings in which chronic HAL exposure had no statistically significant effect on synaptophysin protein levels in the rat or nonhuman primate cortex[34,35,43], by showing HAL did not alter the levels or specific binding of another presynaptic protein, SV2A.

A strength of our clinical study is the use of an in vivo measure of synaptic protein levels, which avoids some potential confounds such as agonal status and cause of death that may affect brain pH and protein and mRNA levels in post-mortem approaches[30]. Moreover, [11C]UCB-J has shown good test–retest reproducibility, indicating it is a reliable imaging tool[44]. Our main outcome measure, $V_T$, is the concentration of the radioligand in the tissue target region relative to that in the plasma at equilibrium and, thus, is proportional to [11C]UCB-J binding to the target in the ROI[45]. It is important to recognise that $V_T$ includes both the concentration of radioligand specifically bound to SV2A and the non-specific uptake (that is, non-specifically bound and free radioligand concentrations). Approximately 80% of [11C]UCB-J $V_T$ in nonhuman primate grey matter regions is accounted for by specific binding[46,47], suggesting the lower values in schizophrenia are likely to be predominantly accounted for by differences in specific binding. To investigate if non-specific uptake was influencing our findings, we repeated the analysis using DVR as the outcome, which uses the centrum semiovale as a pseudoreference region to adjust for non-specific uptake in brain. The centrum semiovale is a white matter region that is largely devoid of SV2A and displays very low uptake of [11C]UCB-J[31,44,48]. Nevertheless, there is a small amount of displacement of [11C]UCB-J uptake in the centrum semiovale by a drug, levetiracetam, that is selective for SV2A[31], suggesting that there is a degree of specific binding in the centrum semiovale. Blocking studies indicate that this is about 8% of the specific binding in grey matter, leading to a slight underestimation of specific binding in grey matter, and white matter may not be an optimal reference region because its tissue composition is different from grey matter[49]. Notwithstanding this, our DVR results indicate that the lower [11C]UCB-J $V_T$ values in schizophrenia likely reflect lower specific binding to SV2A.

Lower GMVs are reported in the FC and ACC in schizophrenia[28], indicating that partial volume effects could contribute to our findings. That we found no significant alterations in regional corrected GMV likely reflects the small-to-medium effect size reductions in volume relative to healthy controls and heterogeneity in the alterations between patients[28], and crucially limits confounding partial volume effects. Furthermore, we did not detect a significant association between regional corrected GMV and regional [11C]UCB-J $V_T$, suggesting that GMVs are unlikely to account for our findings. Interestingly, this also raises the question as to what underlies the widely reported lower GMVs in schizophrenia. In our clinical study, the volunteers are mostly male ($n = 15$ of 18 per group), which may limit generalisations to female patients. However, previous work has not found an effect of sex on [11C]UCB-J $V_T$[50,51]. Six patients were taking clozapine (Supplementary Table 1), which, as this is generally reserved for patients whose illness is antipsychotic treatment resistant[52], could indicate that a third of our sample met criteria for treatment-resistant schizophrenia, although this was not formally assessed. Population studies indicate that antipsychotic treatment resistance is seen in about one-third of chronic and about 20% of first-episode patients[53,54], indicating our sample is representative of chronic schizophrenia but less representative of first-episode patients in this respect. We did not find a significant difference in $V_T$ between clozapine-treated and non-clozapine-treated patients (Supplementary Note 3), although, as the study was not designed to test differences between these groups, it would be interesting to test this in a further study.

It is important to note that increased vesicle number and vesicle clustering effects have been reported in schizophrenia using light microscopy in post-mortem samples[55,56]. However, it is not obvious how such effects would affect our results, given that they do not necessitate alterations in SV2A protein availability and the evidence that synaptic vesicle clustering and disintegration are post-mortem effects[57].

Additional methodological limitations were that all volunteers in the schizophrenia group were taking antipsychotic drug treatment and there were significantly more smokers in the schizophrenia than the control group (Supplementary Table 1; Supplementary Note 1; Supplementary Data 1). Furthermore, although we excluded any volunteers taking any drugs known to bind to SV2A, a small number of patients were taking concomitant psychotropics on top of their antipsychotic medication (Supplementary Note 2; Supplementary Data 1). However, we found no significant association between current antipsychotic dose and [11C]UCB-J $V_T$, and no indication that smoking or concomitant psychotropic treatment influenced $V_T$ (Supplementary Notes 1 and 2). Moreover, a previous study has reported no effect of smoking on [11C]UCB-J $V_T$[51].

Furthermore, data in our preclinical study show that chronic exposure to the antipsychotic drug haloperidol has no significant effects on either SV2A protein levels (as measured by western blot and immunofluorescence), or the specific binding of [3H]UCB-J, a comparable measure to our main neuroimaging end-point in our clinical study. In addition, our data show that SV2A protein levels measured by immunofluorescence are not altered by chronic administration of olanzapine. Strengths of our preclinical study include the combination of three independent but complementary measures of SV2A, and the use of a previously validated model of chronic antipsychotic drug exposure that recapitulates clinic-like plasma levels and pharmacokinetics[36–38]. Antipsychotic drugs accumulate in synaptic vesicles and are secreted upon exocytosis, resulting in increased extracellular drug concentrations during neuronal activity and dose-dependent inhibition of electrically stimulated synaptic vesicle exocytosis[58], which could theoretically affect [11C]UCB-J binding. At least 14 days of HAL treatment, using an identical drug-delivery system to our own work, appears to be necessary for elevated extracellular HAL levels following high potassium stimulation[58]. In the current study, we exposed rats to an identical HAL dose as used in this study, as well as a higher dose of 2 mg/kg/day, for double that time, but did not find any change in SV2A protein or specific binding of [3H]UCB-J. Therefore, whilst additional experiments are required to definitively rule out any effect of antipsychotic-induced vesicle exocytosis on SV2A protein or ligand binding, the aforementioned findings suggest this is unlikely to be the case.

In agreement with the absence of effect of drug administration on SV2A levels detected here, previous studies from our group have consistently found no differences between vehicle and HAL or OLZ exposure in their effects either on rat brain anatomy or at the cellular level[38,59–61]. Notably, our rats were otherwise healthy and so disorder-by-drug interactions were not captured. It has been found previously that, in a rodent model using phencyclidine exposure to recapitulate dopaminergic and cognitive alterations associated with schizophrenia, OLZ treatment rescued exposure-associated synapse loss in the prefrontal cortex but did not affect synaptic density in healthy animals[62]. However, a previous study found no effect of chronic HAL on synaptophysin levels in the hippocampus or FC of a rodent model based on the glutamatergic hypothesis of schizophrenia[63]. We examined the effects of a 28-day exposure (roughly equivalent to 2.4 human years[64], although this is a very simplified model). Most of the volunteers with schizophrenia in our study had taken antipsychotic drugs longer than this. Whilst

the null results in our preclinical study are consistent with those of others examining the effect of HAL on presynaptic protein levels using a similar duration of exposure[34,35,43], our study was not powered to detect small effects so we cannot exclude the possibility that there is a small effect and/or that a longer duration of antipsychotic drug exposure may affect SV2A protein levels or specific binding. Notwithstanding this, our findings from our clinical and preclinical studies suggest that antipsychotic medication exposure is unlikely to account for the lower [11C]UCB-J binding in schizophrenia patients. Nonetheless, future studies investigating SV2A in untreated, first-episode psychosis patients and controlling for smoking would be useful to test this further, and to determine if SV2A changes occur early in the course of illness. It is also important to recognise that our study is only powered to detect a strong relationship with symptoms, so it remains possible that there is a relationship between $V_T$ and symptoms. Longitudinal studies and larger samples are needed to test the time course of alterations and their relationship with symptoms further.

Non-human primate studies show a strong positive relationship between [11C]UCB-J $V_T$ and in vitro SV2A levels determined using western blots ($r > 0.8$) and binding assays ($r > 0.9$)[31]. Moreover, displacement studies[31,47] using levetiracetam, a drug that binds selectively to SV2A[65], show the [11C]UCB-J signal is blocked substantially, indicating specificity to SV2A. This evidence indicates that [11C]UCB-J is a specific marker of SV2A levels, and thus that SV2A levels are reduced in schizophrenia. SV2A, one of three isoforms of SV2, is ubiquitously expressed throughout the brain and is present in GABAergic and glutamatergic presynaptic nerve terminals[66]. Furthermore, SV2A levels are strongly, positively correlated with synaptophysin levels in the brain ($r > 0.95$)[31], which is reduced in disorders associated with synaptic loss[67] and is widely used as a marker of synaptic density. With five copies per synaptic vesicle, SV2 shows much lower variability in this regard than synaptophysin[68]. [11C]UCB-J PET has demonstrated sensitivity to synaptic reductions in temporal lobe epilepsy and Alzheimer's disease[31,69], and has provided evidence consistent with synaptic alterations in affective disorders[51]. Thus, the lower [11C]UCB-J uptake we observe in schizophrenia could be due either to a reduction in SV2A levels specifically, or a reduction in nerve terminal number manifest as decreased levels of SV2A and other synaptic protein levels. SV2A transcript levels have been reported to be lower in the cerebellar cortex in schizophrenia post-mortem[70] but, to our knowledge, there have not been studies in the FC, ACC or hippocampus, and this does not preclude a reduction in synaptic terminal number as well. Moreover, when our findings are taken with evidence of reductions in other synaptic vesicle proteins[8,16,17,41,71], cortical neuropil[72], cortical dendritic spine density[14,15,42] and spine plasticity[73] coupled with unaltered neuronal numbers in schizophrenia[74], the most parsimonious explanation is that they reflect lower synaptic terminal density. A reduction in synaptic terminals could be due to a developmental failure to form synapses and/or excessive synaptic pruning in schizophrenia[7,9], potentially linked to microglial-mediated mechanisms[13,25,75].

Notwithstanding the discussion above on whether our findings reflect a specific loss of SV2A or indicate lower synaptic terminal levels, lower SV2A levels may also have functional consequences. SV2A plays a key role in neurotransmitter exocytosis from vesicles into the synaptic cleft[23]. SV2A knock-out in mice results in deficits in action potential-induced GABAergic neurotransmission[76], and pharmacological modulation of SV2A function alters post-synaptic excitatory and inhibitory potentials[77]. Moreover, SV2A levels in the dorsolateral prefrontal cortex indexed using [11C]UCB-J are negatively associated with functional connectivity between that region and the posterior cingulate cortex in patients with affective disorders[51]. Thus, lower SV2A levels could have functional

consequences, theoretically altering excitatory–inhibitory balance, thereby impairing interactions between neuronal systems and contributing consequently to the dysconnectivity observed in schizophrenia[78], although this requires testing.

Finally, our finding that there is no association between symptom severity and [11C]UCB-J $V_T$ could suggest that synaptic alterations occur early in the course of the illness, consistent with models that early neurodevelopment is disrupted in schizophrenia with subsequent changes in dopamine and other neurotransmitter systems leading to psychotic symptoms[75,79].

In conclusion, SV2A levels are significantly lower in the FC and ACC in schizophrenia, and antipsychotic drug exposure at clinically relevant doses does not significantly alter SV2A levels or [3H]UCB-J-specific binding in the prefrontal and cingulate cortices of naïve rats. These findings indicate that synaptic alterations occurs in vivo in schizophrenia, and lower SV2A levels are unlikely to be directly accounted for by antipsychotic drug treatment.

## Methods

**Schizophrenia patient and healthy volunteer recruitment**. The study protocol (Integrated Research Application System reference number: 209761) was approved by the London-West London & GTAC Research Ethics Committee (study reference: 16/LO/1941) and approval to administer radioactive material was obtained from the Administration of Radioactive Substances Advisory Committee (ARSAC, UK). All human participants provided written, informed consent before participating in the study, which was conducted in accordance with the Declaration of Helsinki.

We recruited 18 individuals with a DSM-5 diagnosis of schizophrenia from community mental health services in London. Eighteen healthy volunteers were recruited through public advertisement. Inclusion criteria for all volunteers were: aged between 18 and 65 years old, demonstrated capacity to consent, and had a normal blood coagulation test.

In addition, individuals with schizophrenia were required to meet DSM-5 criteria for a diagnosis of schizophrenia and to have had no changes in treatment for at least 4 weeks prior to the screening visit; and healthy volunteers were required to have no history of a current or lifetime diagnosis of a mental disorder or family history of schizophrenia.

Exclusion criteria for all volunteers were: history of neurological disorder, head trauma resulting in a loss of consciousness, drug or alcohol dependence (except for nicotine dependence); significant medical disorder; taking a drug known to interact with SV2A (including levetiracetam, brivaracetam, loratadine or quinine);[80] or contraindications to imaging.

**Clinical assessments**. Volunteers with schizophrenia underwent a psychiatric symptom assessment by a psychiatrist using the PANSS[81] to assess symptom severity and the Structured Clinical Interview for DSM-5 to confirm the diagnosis and to assess for psychiatric co-morbidities. Healthy volunteers were screened with the Structured Clinical Interview for DSM-5 to exclude any psychiatric illness. The healthy volunteers were screened to exclude any family history of psychosis.

**Magnetic resonance imaging**. All subjects underwent structural magnetic resonance imaging (MRI) to facilitate the anatomical delineation of ROIs. For all patients and 16 of the healthy volunteers, T1-weighted three-dimension magnetisation-prepared rapid acquisition gradient echo (MPRAGE) images were acquired on a Siemens Magnetom Prisma 3T scanner (Siemens, Erlangen, Germany) according to the following parameters: repetition time = 2300.0 ms, echo time = 2.28 ms, flip angle = 9°, field of view (FOV) = 256 × 256 mm, 176 sagittal slices of 1-mm thickness, distance factor = 50%, voxel size = 1.0 × 1.0 × 1.0 mm. For two healthy volunteers, T1-weighted three-dimension MPRAGE images were acquired on a Siemens 3T Trio clinical MRI scanner (Siemens Healthineers, Erlangen, Germany) with the following parameters: repetition time = 2300.0 ms, echo time = 2.98 ms, flip angle = 9°, FOV = 256 × 256 mm, 160 sagittal slices of 1-mm thickness, distance factor = 50%, voxel size = 1.0 × 1.0 × 1.0 mm.

**PET imaging**. All participants underwent a PET scan with [11C]UCB-J, a radioligand specific for SV2A[31,47]. Each subject had a CT scan for attenuation correction purposes 2 min prior to the injection of radiotracer. The study physician administered a microdose of [11C]UCB-J (target activity approximately 300 MBq) via an intravenous cannula as a smooth bolus injection over 20 s. PET data were acquired for 90 min using a Biograph 6 HiRez PET-CT scanner (Siemens, Erlangen, Germany). Arterial blood was collected throughout the PET scan unilaterally via the radial artery to measure the arterial input function.

**Arterial blood sampling**. Whole-blood activity was measured using a continuous automatic blood sampling system (Allogg AB, Mariefred, Sweden). Discrete samples were taken at 10, 15, 20, 25, 30, 40, 50, 60, 70, 80 and 90 min after tracer injection. Total radioactivity concentration was evaluated in blood and plasma using a Perkin Elmer 1470 10-well gamma counter. Discrete blood samples were used to determine the plasma radioactivity fraction constituted by unchanged parent radioligand using high-performance liquid chromatograph analysis. The [11C]UCB-J plasma-free fraction was measured by ultrafiltration in triplicate using an arterial blood sample taken prior to tracer injection.

**Image analysis**. Processing and modelling were conducted using MIAKAT software version 4.3.7 (http://www.miakat.org/MIAKAT2/index.html), which implements functions from MATLAB (Mathworks Inc., Natick, MA, USA), FSL (version 5.0.10; FMRIB, Oxford, UK) and Statistical and Parametric Mapping12 (SPM12, Wellcome Trust Centre for Neuroimaging, http://www.fil.ion.ucl.ac.uk/spm).

Each subject's MRI underwent brain extraction using FSL, and grey matter segmentation and rigid body coregistration to a standard reference space[82] using SPM12 as implemented via MIAKAT. The template brain image and associated Clinical Imaging Centre (CIC) atlas[83] were then warped nonlinearly to the individual subject's MRI image where the FC, ACC and hippocampus were defined as the primary ROIs (Supplementary Fig. 6). The CIC atlas was also used to define the dorsolateral prefrontal cortex, temporal, parietal and occipital lobes, thalamus and amygdala as additional ROIs for the exploratory analysis of effects in other brain regions (Supplementary Fig. 6). The centrum semiovale ROI was generated from the automated anatomical labelling template[84] according to parameters defined for its use as a reference region for nondisplaceable binding of [11C]UCB-J[31] (Supplementary Fig. 6).

PET images were registered to each subject's MRI image and corrected for motion using frame-to-frame rigid-body registration, with the 14th frame (acquired 540–660 s into the scan) as the reference frame as this shows good anatomical delineation. Regional time activity curves (TACs) were generated for each ROI.

Regional TAC and arterial input function data were analysed together using the one-tissue compartment model (1TCM), which has been shown to produce reliable estimates of [11C]UCB-J volumes of distribution ($V_T$)[44,48].

Grey matter masks were applied to the ROIs within MIAKAT to extract both the grey matter $V_T$ and the GMV of the ROI. For each subject, regional corrected GMV is expressed as proportions of the subject's own total brain volume, including grey matter, white matter and cerebrospinal fluid, to account for intersubject variation in total brain volume. Furthermore, regional DVR was obtained by use of the centrum semiovale as a pseudoreference region[31,48].

**Animals and antipsychotic drug administration**. Animal experiments were carried out in accordance with the Home Office Animals (Scientific Procedures) Act (1986) and European Union (EU) Directive 2010/63/EU, with the approval of the local Animal Welfare and Ethical Review Body (AWERB) panel at King's College London (KCL). Male Sprague-Dawley rats (Charles River UK Ltd, Margate, UK), initial body weight 240–270 g (6–10 weeks of age), were administered with vehicle (β-hydroxypropylcyclodextrin, 20% wt/vol, acidified by ascorbic acid to pH 6), HAL (0.5 or 2 mg/kg/day; Sigma-Aldrich, Gillingham, UK) or OLZ (7.5 mg/kg/day; Sigma-Aldrich, Gillingham, UK) for 28 days (~2.4 human years based on 11.8 rat days as equivalent to 1 human year[64]) using osmotic minipumps (Alzet Model 2ML4; 28 days; Alzet, Cupertino, CA, USA) inserted subcutaneously on the back flank. After 28 days, animals were terminally anaesthetised by injection of sodium pentobarbital (60 mg/kg, intraperitoneal) and culled by cardiac perfusion using heparinised ice-cold 0.9% saline[37,38]. A blood sample was collected at termination for estimation of drug levels, measured using tandem mass spectrometry (Cyprotex, Macclesfield, UK). Three cohorts of vehicle and drug-exposed rats were generated for different experimental end-points. Specifically, cohort 1 comprised vehicle- (n = 10); 0.5 mg/kg/day (n = 12) or 2 mg/kg/day HAL- (n = 11) treated groups; following perfusion, brain tissue was snap-frozen to be utilised for western blotting or, in a subset of these animals, for [3H]UCB-J autoradiography (vehicle, n = 7; 0.5 mg/kg/day HAL, n = 3; 2 mg/kg/day HAL, n = 5). Cohort 2 comprised vehicle (n = 11) and 0.5 mg/kg/day HAL- (n = 11) exposed rats; brain tissue was drop-fixed overnight in 4% PFA at 4 °C, and then cryoprotected in buffered 30% sucrose solution for 48 h at 4 °C prior to freezing and processing for post-mortem analysis of SV2A intensity using immunofluorescence staining and confocal microscopy. Cohort 3 comprised vehicle- (n = 4) and 7.5 mg/kg/day OLZ- (n = 12) exposed rats. Brain tissue was fixed and processed for immunostaining in the same way as brain tissue from cohort 2.

**Synaptosome extraction and western blotting**. The frontal cortices (FC—prefrontal and cingulate cortex) of snap-frozen hemispheres were dissected on ice and homogenised in lysis buffer (150 mM NaCl, 1 mM EDTA, 1 mM EGTA, 20 mM Tris pH 7.4, 1% Triton X-100, supplemented with a cocktail of protease and phosphatase inhibitors (Pierce, UK)) using a plastic pestle. Total lysate was centrifuged for 8 min at 2000 rpm at 4 °C to remove nuclei. The resultant supernatants were then centrifuged for 15 min at 13,000 rpm at 4 °C. The resultant pellet containing crude synaptosomes was resuspended in lysis buffer. A total of 30 μg of protein from each sample was analysed by western blots probed with rabbit polyclonal α-SV2A (ab32942, Abcam, Cambridge, UK; 1:2000) or mouse

monoclonal α-GAPDH (60004-1, Protein Tech, 1:10,000) and the appropriate secondary antibodies. Protein bands were visualised using the Biorad ChemiDoc scanner using Image LabTM Software (PC Version 6.0 SOFT-LIT-170-9690-ILSPC) and quantified using Image Studio (Version 5.2.5, Li-Cor Biosciences). SV2A protein intensity was normalised to that of GAPDH as the loading control for each rat. See Supplementary Fig. 4 for unprocessed western blot images of SV2A and GAPDH protein levels in synaptosome fractions purified from the rat FC.

**Ex vivo autoradiography for SV2A using [³H]UCB-J**. Coronal sections (20-µm-thick) from a subset of the hemispheres from cohort 1 were thaw mounted on 1% gelatinised superfrost slides. Individual slides containing 10–12 adjacent tissue sections at 500 µm intervals were then preincubated in assay buffer (50 mM Tris Base, 140 mM NaCl, 1.5 mM MgCl₂, 5 mM KCl, 1.5 mM CaCl₂, pH 7.4) for 10 min at room temperature (RT), and subsequently incubated for 2 h at RT in assay buffer containing either [³H]UCB-J, a radioligand specific for SV2A[85] (12.5 nM), or [³H]UCB-J in the presence of levetiracetam (Sigma, Gillingham, UK; diluted to 1 mM in DMSO) to assess total and non-specific binding, respectively. After incubation, slides were washed twice in ice-cold wash buffer (50 mM Tris Base, 1.4 mM MgCl₂, pH 7.4), followed by a brief rinse in ice-cold reverse osmosis (RO) water and dried in a cool airstream. Sections were apposed to Carestream Kodak Biomax Light film (Sigma-Aldrich) adjacent to tissue-equivalent tritium microscale standards (American Radiolabeled Chemicals) for 6 weeks before the autoradiograms were developed and analysed using MCID Basic 7.0 software (Interfocus, Cambridge, UK).

**Fluorescence immunostaining**. Coronal sections (30-µm thick, interval 1/12, 360 µm spacing between sections) from one hemisphere of each animal in cohorts 2 and 3 were stained free-floating. For antigen retrieval, sections were incubated for 10–15 min in 10 mM sodium citrate (pH 6.2) at RT, followed by incubation in pre-heated 10 mM sodium citrate (pH 6.2) in a water-bath at 78 °C. Sections were then allowed to cool down to RT in the same solution while gently shaking for 30 min. After a brief wash in PBS supplemented with 0.05% Triton X-100, sections were incubated for 4 h in blocking solution (10% NGS, 1.5% BSA, 0.3% Triton X-100 in PBS) and then overnight with primary antibody (Rabbit-α-SV2A, Abcam ab32942; 1:1000). The next day, sections were incubated for 2 h in secondary antibody solution (Goat-α-rabbit Alexa-555, abcam ab150090; both 1:1000), mounted on Superfrost Plus slides (Thermo Fisher), and air-dried at RT for 1 h before cover-slipping with mounting medium containing DAPI (Vectashield).

**Confocal image acquisition and analysis**. Images of SV2A staining were acquired as a stack with an interval of 0.3 µm using an Inverted Spinning Disk confocal microscope (Nikon, JP) and a 60× oil immersion lens objective (NA 1.4). SV2A intensity was analysed using an in-house written macro in ImageJ (https://imagej.net/Welcome). Total SV2A staining intensities were measured from three consecutive optical sections within each image stack, selected based on quality of staining and contrast in the image, to ensure proper synaptic staining. These optical sections were then combined by maximum intensity projection. Total SV2A intensity was measured after background subtraction using a rolling ball with radius of 25 pixels (5 µm) to remove diffuse (and presumably non-synaptic) staining.

See Supplementary methods for further information.

**Statistical analysis of [¹¹C]UCB-J PET data**. Statistical analyses were performed using GraphPad Prism version 8.00 for Mac (GraphPad Software, La Jolla, California, USA, (www.graphpad.com)), IBM SPSS Statistics, Version 25, and RStudio Version 1.1.456 (RStudio Team (2016), RStudio, Inc., Boston, MA (http://www.rstudio.com/)). We tested for normality of distribution using the Shapiro–Wilk test. A two-way analysis of variance (ANOVA) was used to test the effects of group and ROI, and group-by-ROI interactions, on [¹¹C]UCB-J $V_T$ and to explore if differences in corrected GMV contributed to our $V_T$ findings. Where there were significant effects, planned post-hoc independent samples $t$-tests (two-tailed) were used to test the effect of group on $V_T$ and corrected GMV at each ROI. For post-hoc analyses, an FDR (Q) of 5% was used to limit false discoveries when performing multiple comparisons between groups in the three ROIs[86] and FDR-adjusted $p$ values are reported. Group differences in clinico-demographic variables were assessed using independent sample $t$-tests for normally distributed data, Fisher's test for categorical data and Kolmogorov–Smirnov tests for nonparametric data. We tested if there were significant associations between grey matter SV2A levels and GMV, PANSS scores, antipsychotic dose and duration of illness using the Pearson product-moment correlation coefficient for normally distributed data, and Spearman's rank correlation for non-normally distributed data. The relationship with antipsychotic dose was explored using chlorpromazine-equivalents derived using the defined daily doses (DDD) method[87].

We conducted an exploratory analysis of other brain regions to assess whether there were significant alterations in [¹¹C]UCB-J $V_T$ in other brain regions and in [¹¹C]UCB-J DVR in schizophrenia using two-way ANOVA with planned post-hoc independent samples $t$-tests with FDR correction for multiple comparisons.

**Statistical analyses of animal data**. Statistical analyses were performed using Prism version 8.0.0 (GraphPad Software, La Jolla, California, USA). Data were tested for a normal distribution using the Shapiro-Wilk test. Western blot data were analysed using non-parametric Kruskal–Wallis test, with $\alpha = 0.05$. Auto-radiography and SV2A immunostaining data were analysed by two-way ANOVA with Bonferroni's post-hoc test. Researchers were blinded to the treatment groups for all analyses.

**Reporting summary**. Further information on research design is available in the Nature Research Reporting Summary linked to this article.

## Data availability

The imaging and related clinical data have been deposited on the NODE PET data repository (listed as SV2A imaging in schizophrenia) and are available at: https://molecular-neuroimaging.com/neuroimaging-database/.

## Code availability

Code generated and used in the production of this manuscript is available on reasonable request.

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

## Acknowledgements
The authors thank Ben Statton, Marina Quinlan, Alaine Berry, Rohini Akosa, Ryan Janisch, Daniela Ribeiro and Jim Anscombe for their expert assistance. ODH and ACV acknowledge financial support for this study from the Medical Research Council (grant nos. MC-A656-5QD30 and MR/L022176/1), Wellcome Trust (no. 094849/Z/10/Z) and the National Institute for Health Research (NIHR) Biomedical Research Centre at South London and Maudsley NHS Foundation Trust and King's College London.

## Author contributions
Conceptualisation: E.C.O., E.F.H., S.N., A.C.V. and O.D.H. Data curation: E.C.O. and E.F.H. Formal analysis: E.C.O., E.F.H., A.C.V. and O.D.H. Funding acquisition: T.R.M., A.C.V. and O.D.H. Investigation: E.C.O., E.F.H., T.W., M.C.C., H.C., D.B., M.R. and E.S. Methodology: E.C.O., E.F.H., A.M., A.C.V. and O.D.H. Project administration: E.C.O., S.N. and O.D.H. Resources: M.C.C., L.W. and O.D.H. Software: A.M. Supervision: L.W., A.C.V. and O.D.H. Validation: A.M., E.A.R. and R.N.G. Visualisation: E.C.O. and E.F.H. Writing—original draft: E.C.O. and E.F.H. Writing—review and editing: all authors.

## Competing interests
A.M., L.W., D.B., E.A.R. and R.N.G. are employees of Invicro LLC. R.N.G. is a consultant for AbbVie, Biogen & Cerveau. O.D.H. has received investigator-initiated research funding from and/or participated in advisory/speaker meetings organised by Angellini, Astra-Zeneca, Autifony, Biogen, BMS, Eli Lilly, Heptares, Jansenn, Lundbeck, Lyden-Delta, Otsuka, Servier, Sunovion, Rand and Roche. Neither O.D.H. nor his family have been employed by or have holdings/a financial stake in any pharmaceutical company. The other authors declare no competing interests.
