## [Peer Review File · Nature Communications]

Reviewers' comments:

Reviewer #1 (Remarks to the Author):

This is an interesting study looking at a synaptic PET marker (SV2A) in patients with schizophrenia and controls. This is important as genetic studies have highlighted synaptic processes in schizophrenia (and related disorders). So this provides a pathological measure. They also pair this with an animal study in an attempt to show that the observed effects are not simply due to medication. This is an important addition. I have a few comments:

1. The authors correctly highlight that genetic studies in schizophrenia point to synaptic pathways. However the claim that there is a specific link to SV2A is not confirmed in substantive studies and should be removed.
2. They have controlled for medication effects in the animal study. Can they provide reassurance that there are no correlations of the signal with other potential confounds eg smoking, other medications apart from antipsychotics etc? (the latter should be listed).
3. There are no correlations with Grey matter- have the authors examined correlations with measures of white matter?
4. What is known in terms of how SV2A levels relate to synaptic function overall? What aspects is being assessed here? Is it "number of synapses" or specific aspects of function - this could be made clearer.
5. Could the lack of significant findings in the Hippocampus represent the quality of the signal from this brain region?

Reviewer #2 (Remarks to the Author):

In this interesting manuscript the authors report the first study on SV2A PET imaging in patients with schizophrenia. The study was performed by researchers which have significant experience in the field of schizophrenia and brain imaging. The experiments took a translational approach, including an evaluation of potential anti-psychotic treatment effects on SV2A examined in animals. The sample size in the clinical study (16 schizophrenia and 16 control volunteers) and the animal study (n=3-12 per group) were of reasonable size for these type of study designs. The imaging and biochemical studies were performed with appropriate methodologies and were reported in good detail. Overall, the study was of high quality and provides important insight in in vivo SV2A changes in schizophrenia (although no correlations were found with clinical symptoms).

My main comments:

- 1) A more detailed summary of the studied subjects (e.g. clinical assessment details) should be added. For instance, there was no group effect on the corrected grey matter volumes (cGMV), which is different than several reports in the literature. Could this be related to the particular group of schizophrenia volunteers studies, could they be due to methodological differences (e.g. use of FreeSurfer) or are they related to something else?
- 2) Vt was used as the main outcome measure in the PET studies. It has however been shown that the white matter contains minimal SV2A, and that the centrum semiovale may be used as a reference region for [11C]UCB-J (Koole et al, 2019, EJNMI). Since Vt typically has larger inter-subject variability when compared to DVR-1, did the authors consider using a reference region approach for quantification of [11C]UCB-J? How did the Vt values of the white matter compare? It would be helpful for the reader if these values were included.
- 3) SV2A plays a role in exocytosis, and the effect of exocytosis on the binding of [11C]UCB-J

seems so far unclear. In the current study design, the animal study does not control for a D2R drug treatment effect on exocytosis and potential following effect on in vivo [11C]UCB-J binding. The effect of exocytosis on [11C]UCB-J binding should thus also be discussed as another possible reason for the group difference in V_t .

Additional comments:

- 1) How were the secondary ROIs selected?
- 2) Where the groups matched for smoking status?
- 3) Figure 2, in the Western blot results the ratio of SV2A/GAPDH is relatively low, 0.05-0.1. Did the authors also compare the uncorrected SV2A intensities?
- 4) Line 296, the V_t value also includes the free radiotracer concentration, in addition to the specific and nonspecific binding.
- 5) Line 297, the proportion of V_t reflecting specific binding relates to the regional SV2A density. In which brain region does 80% of V_t account for specific binding?
- 6) Line 353, should also refer to the loss of SV2A in MDD, reference 76.
- 7) Line 480, what time window corresponds to the 14th frame?
- 8) Line 486, where the brain volumes determined using FreeSurfer?
- 9) The authors interpret SV2A binding as a marker of synaptic proteins and as a proxy marker for nerve terminals. It is not discussed how SV2A is distributed across neurons from different neurotransmitters (e.g. glutamatergic vs. Gaba-ergic), and the authors should discuss this in further detail.

Reviewer #3 (Remarks to the Author):

Onwordi and colleagues present the findings of two complementary studies supporting the synaptic hypothesis in schizophrenia. In the clinical study, the authors examined 16 patients with schizophrenia and 16 healthy controls using a novel PET ligand for synaptic vesicle glycoprotein 2A (SV2A). To control for the possible effects of antipsychotic treatment, the authors performed a parallel animal study. They investigated the effects of haloperidol and olanzapine on SV2A levels in Sprague-Dawley rats, showing no significant effects. Though this is a relatively small study, the investigators present compelling in vivo evidence for the presence of decreased levels of a synaptic protein in frontal and anterior cingulate cortices in the brains of patients with schizophrenia and show that this is likely not caused by the antipsychotic treatment, but rather the disease itself. This study addresses an important question (whether patients with schizophrenia exhibit decreased levels of synaptic proteins) using a novel imaging technique. This report is an important addition to the literature, with the following suggestions for improvement.

1. Power analyses should be presented for the human and rat studies. This is important because, at least for the human experiment, null findings (e.g., no significant differences in binding in the hippocampus) do not provide strong evidence of no group difference. More specifically, a sample size of 16 per group is only powered to detect large effect sizes. (For example, a between-group comparison using a two-sample t test has ~80% power to detect $d=1.0$ with $\alpha=0.05$.) Correlation tests of clinical variables with $n = 16$ (Table 2) are similarly limited. This limitation should be addressed in the Discussion.

2. Related to this point, "regional specificity" (line 272) is supported by the significant group-by-ROI interaction (line 115), but these results only provide evidence that the effect size for group differences in hippocampus are not "large" (lines 272-273). Indeed, Figure 1 and the p value of 0.13 suggest that the effect size in hippocampus could be small or moderate.

3. Please provide images showing the a priori and exploratory regions of interest used in the human study.

4. Please show individual data points in Figure 1.
5. Please show effect sizes with p values, even those for which $p > 0.05$, in Figure 1 and Supplementary Figures 1 and 2.
6. Please provide more clinical descriptive details. This is important for reproducibility, since clinical features of patient samples can vary widely, and those clinical features may be associated with biological measures like synaptic density. I suggest at least the following clinical variables should be described, if available: PANSS sub-scales, other concomitant medications (besides antipsychotics), age of onset of psychosis, family history of psychosis, duration of illness, nicotine/smoking status, co-morbid DSM diagnoses (anxiety, depression, OCD, past substance use disorder, etc).
7. If previous studies have validated the specificity of the SV2A polyclonal antibody and the UCB-J ligand, please cite them.
8. Nicotine use often differs between schizophrenia and control samples. Please report nicotine/tobacco use for the two groups and, if needed, address this potential confounder in the Discussion.
9. The groups are well matched in terms of age and gender. However, both groups are predominantly male. Were sampling procedures biased toward men? Are sex differences expected in the variables measured? Are UCB-J PET data available comparing women vs men? How does this imbalance affect interpretation and generalizability? This limitation should be stated in the Abstract and addressed in the Discussion.
10. Line 49 – I would recommend specifying "positive" symptoms, in contrast to negative symptoms, rather than the formulation "psychotic, cognitive, and negative."
11. Line 69-71 – The authors cite studies showing lower brain volumes and altered functional connectivity in schizophrenia as indirect evidence for altered synaptic function, but there many other potential explanations for these findings, e.g., excitotoxicity/ medication effects accounting for lower volumes, white matter changes leading to altered anatomical connectivity, leading in turn to altered functional connectivity.
12. Line 81 - The term "synaptic dysfunction" is used in several places in the manuscript. Since the experiments presented do not measure SV2A or brain function, I would advise using this term with care. While the data do support the "synaptic dysfunction" hypothesis (Abstract), these experiments do not strongly or directly test differences in function (line 81).
13. Line 83 – Frontal cortex, ACC, and hippocampus are all paired structures. Did the authors hypothesize any lateralized effects?
14. Line 85 - The meaning of "indirectly associated" is unclear.
15. Line 108 – The authors state that "all volunteers with schizophrenia were on dopamine D2 receptor antagonist medication". This is technically not true, since one patient is on a partial agonist, aripiprazole. It might be better to use a more general term, such as "antipsychotics".
16. Line 185 – How does the length of exposure of rats – 28 days (elsewhere in the text, the authors point out this is roughly equivalent to 2.4 human years) – compare to the length of time the human subjects have been exposed to antipsychotics?
17. Lines 421-422 – Please specify what is meant by "schizophrenia spectrum disorders"

18. Line 431 – Were volunteers with schizophrenia diagnosed with the SCID, or was it a clinical diagnosis? Who performed the assessments and administered the scales? Psychiatrists? Non-MD raters? One or multiple? If there were more than one rater, did the authors measure inter-rater reliability?

19. Line 777 (Supplemental Table 1) -- 5 of the 16 patients were on clozapine, which raises questions about treatment resistance, and duration of illness. Did treatment resistant patients or patients on clozapine have different VT than the non-treatment resistant ones?

Dear Reviewers,

Thank you for your review of our manuscript.

The manuscript has been revised to incorporate all of the suggestions put forward.

We have addressed the points that you have raised, with our itemized response attached herewith. We hope the revised version will meet your approval.

Yours sincerely,

The authors.

Reviewers' comments:

Reviewer #1 (Remarks to the Author):

This is an interesting study looking at a synaptic PET marker (SV2A) in patients with schizophrenia and controls. This is important as genetic studies have highlighted synaptic processes in schizophrenia (and related disorders). So this provides a pathological measure. They also pair this with an animal study in an attempt to show that the observed effects are not simply due to medication. This is an important addition. I have a few comments:

1. The authors correctly highlight that genetic studies in schizophrenia point to synaptic pathways. However the claim that there is a specific link to SV2A is not confirmed in substantive studies and should be removed.

>> We thank the reviewer for this comment. Multiple studies report variants in genes encoding synaptic proteins in schizophrenia. In support of this, we referred to one study that identified a single nucleotide polymorphism in *SV2A* associated with schizophrenia¹. It is the case, however, that this finding has not been replicated, nor has it been identified in the GWAS data. In light of this, we have qualified our claim such that the text in the introduction now reads as follows:

Genetic studies have found associations between schizophrenia and variants in genes encoding synaptic proteins²⁻⁴. A polymorphism in the synaptic vesicle glycoprotein 2A gene (*SV2A*) has been associated with increased schizophrenia risk¹, although this finding has not

been replicated in genome wide association studies. Associations between schizophrenia and variants in genes shown to mediate synaptic pruning in postnatal mouse neurodevelopment have been reported ⁵.

2. They have controlled for medication effects in the animal study. Can they provide reassurance that there are no correlations of the signal with other potential confounds eg smoking, other medications apart from antipsychotics etc? (the latter should be listed).

>>We thank the reviewer for their considerations regarding the potential effects of smoking and other medications. We have added in additional information on smoking and concomitant medications, conducted additional analyses, and amended the discussion to address this as follows:

Amendment to the results section:

We conducted exploratory analyses to explore the potential influence of smoking, concomitant medication and clozapine treatment on V_T . These showed no significant effect of these variables on V_T (see Supplementary Notes).

Supplementary Notes:

A significantly greater proportion of the schizophrenia group were current smokers as compared to the healthy volunteer group ($p = 0.002$, two-tailed Fisher's exact test). In view of this we conducted an exploratory analysis of the effect of smoking status on [¹¹C]UCB-J

V_T . This showed there was no significant effect of smoking ($F_{1,16} = 1.50$, $p=0.24$) nor smoking-by-ROI interaction ($F_{1.5,23.7} = 0.68$, $p = 0.48$) on [^{11}C]UCB-J V_T .

Effect of taking concomitant psychotropic medications in schizophrenia group on [^{11}C]UCB-J V_T in a priori ROIs

Five of the schizophrenia patients were taking other psychotropic drugs in addition to antipsychotics (Supplementary Table 2). Although none of these drugs are known to bind to SV2A, we conducted an exploratory analysis in case there was an indirect effect of concomitant medication on [^{11}C]UCB-J V_T . This showed there was no significant effect of concomitant psychotropic medication ($F_{1,16} = 1.58$, $p=0.23$) nor concomitant psychotropic medication-by-ROI interaction ($F_{2,32} = 0.20$, $p = 0.82$) on [^{11}C]UCB-J V_T . Post hoc analyses with false-discovery rate (FDR) adjustment revealed that mean (SEM) [^{11}C]UCB-J V_T (ml/cm^3) was not significantly altered in those taking concomitant psychotropic medications (CPM) relative those not taking concomitant psychotropic medications (nCPM) the FC (nCPM = 17.50 [0.82]; CPM = 15.46 [1.95]; $t=1.15$, $df=16.0$, $p = 0.27$), ACC (nCPM = 20.13 [0.64]; CPM = 18.06 [2.15]; $t=1.26$, $df=16.0$, $p = 0.27$) or hippocampus (nCPM = 14.55 [0.63]; CPM = 12.89 [1.35]; $t=1.27$, $df=16.0$, $p = 0.27$).

Amendment to the discussion:

Additional methodological limitations were that all volunteers in the schizophrenia group were taking antipsychotic drug treatment and there were significantly more smokers in the schizophrenia than the control group (Supplementary Notes; Supplementary Table 2).

Furthermore, although we excluded any volunteers taking any drugs known to bind to SV2A, a small number of patients were taking concomitant psychotropics on top of their antipsychotic medication (Supplementary Notes; Supplementary Table 2). However, we found no significant association between current antipsychotic dose and [¹¹C]UCB-J V_T , and no indication that smoking or concomitant psychotropic treatment influenced V_T (Supplementary Notes). Moreover, a previous study has reported no effect of smoking on [¹¹C]UCB-J V_T ⁶. Nonetheless, future studies investigating SV2A in untreated, first-episode psychosis patients and controlling for smoking would be useful to test this further, and determine if SV2A changes occur early in the course of illness.

3. There are no correlations with Grey matter- have the authors examined correlations with measures of white matter?

>> We did not correlate V_T with white matter because negligible amounts of SV2A are found in white matter regions, which display correspondingly low uptake of [¹¹C]UCB-J V_T relative to grey matter regions⁷⁻⁹. Furthermore, data from our preclinical study confirmed the lack of [³H]UCB-J binding and absence of SV2A immunostaining in the white matter (Figures 4C and E). Thus we believe that this analysis would not be informative.

4. What is known in terms of how SV2A levels relate to synaptic function overall?

>> We thank the reviewer for this question, and have updated the discussion section to reflect our answer as follows:

Notwithstanding the discussion above on whether our findings reflect a specific loss of SV2A or indicate lower synaptic terminal levels, lower SV2A levels may also have functional consequences. SV2A plays a key role in neurotransmitter exocytosis from vesicles into the synaptic cleft¹⁰. SV2A knock-out in mice results in deficits in action potential-induced GABAergic neurotransmission¹¹, and pharmacological modulation of SV2A function alters post-synaptic excitatory and inhibitory potentials¹². Moreover, SV2A levels in the dorsolateral prefrontal cortex indexed using [¹¹C]UCB-J are negatively associated with functional connectivity between that region and the posterior cingulate cortex in patients with affective disorders⁶. Thus, lower SV2A levels could have functional consequences, theoretically altering excitatory-inhibitory balance, thereby impairing interactions between neuronal systems and contributing consequently to the dysconnectivity observed in schizophrenia^{13,14}, although this requires testing.

5.What aspects is being assessed here? Is it "number of synapses" or specific aspects of function - this could be made clearer.

>> We assessed levels of SV2A, which is both a marker of synapses, but also has a functional role. To address the reviewer's comment, we have elaborated in the discussion of the revised version on the relationship between SV2A levels and synaptic density vs synaptic function as requested, as shown below:

Non-human primate studies show a strong positive relationship between [¹¹C]UCB-J V_T and *in vitro* SV2A levels determined using western blots ($r>0.8$) and binding assays ($r>0.9$)⁷. Moreover, displacement studies^{7,9,15} using levetiracetam, a drug that binds selectively to SV2A¹⁶, show the [¹¹C]UCB-J signal is blocked substantially, indicating specificity to SV2A. This evidence indicates that [¹¹C]UCB-J is a specific marker of SV2A levels, and thus

that SV2A levels are reduced in schizophrenia. SV2A, one of 3 isoforms of SV2, is ubiquitously expressed throughout the brain and is present in GABAergic and glutamatergic presynaptic nerve terminals¹⁷. Furthermore, SV2A levels are strongly, positively correlated with synaptophysin levels in the brain ($r > 0.95$)⁷, which is reduced in disorders associated with synaptic loss^{7,18} and is widely used as a marker of synaptic density. With 5 copies per synaptic vesicle, SV2 shows much lower variability in this regard than synaptophysin¹⁹. [¹¹C]UCB-J PET has demonstrated sensitivity to synaptic reductions in temporal lobe epilepsy and Alzheimer's disease^{7,18}, and has provided evidence consistent with synaptic alterations in affective disorders⁶. Thus, the lower [¹¹C]UCB-J uptake we observe in schizophrenia could be due either to a reduction in SV2A levels specifically, or a reduction in nerve terminal number manifest as decreased levels of SV2A and other synaptic protein levels. SV2A transcript levels have been reported to be lower in the cerebellar cortex in schizophrenia *post-mortem*²⁰ but, to our knowledge, there have not been studies in the FC, ACC or hippocampus and this does not preclude a reduction in synaptic terminal number as well. Moreover, when our findings are taken with evidence of reductions in other synaptic vesicle proteins²¹⁻²⁶, cortical neuropil²⁷, cortical dendritic spine density²⁸⁻³⁰, and spine plasticity³¹ coupled with unaltered neuronal numbers in schizophrenia³², the most parsimonious explanation is that they reflect lower synaptic terminal density. A reduction in synaptic terminals could be due to a developmental failure to form synapses and/or excess synaptic pruning in schizophrenia^{33,34}, potentially linked to microglial mediated mechanisms^{5,35,36}.

Notwithstanding the discussion above on whether our findings reflect a specific loss of SV2A or indicate lower synaptic terminal levels, lower SV2A levels may also have functional consequences. SV2A plays a key role in neurotransmitter exocytosis from vesicles into the synaptic cleft¹⁰. SV2A knock-out in mice results in deficits in action potential-induced

GABAergic neurotransmission¹¹, and pharmacological modulation of SV2A function alters post-synaptic excitatory and inhibitory potentials¹². Moreover, SV2A levels in the dorsolateral prefrontal cortex indexed using [¹¹C]UCB-J are negatively associated with functional connectivity between that region and the posterior cingulate cortex in patients with affective disorders⁶. Thus, lower SV2A levels could have functional consequences, theoretically altering excitatory-inhibitory balance, thereby impairing interactions between neuronal systems and contributing consequently to the dysconnectivity observed in schizophrenia^{13,14}, although this requires testing.

6. Could the lack of significant findings in the Hippocampus represent the quality of the signal from this brain region?

>> We thank the reviewer for this comment. It is certainly possible that significant differences in [¹¹C]UCB-J V_T may be harder to detect in the hippocampus because the specific signal is lower compared to those of the frontal and anterior cingulate cortices.

To address this point, we have now amended the discussion to consider this issue and have also conducted additional analyses using distribution volume ratio (DVR), which has a higher signal to noise ratio than V_T (see reviewer 2 point 3), and so is more sensitive to differences. These additional analyses do show a significant difference in the hippocampus, consistent with the reviewer's point that the lack of significant difference with V_T could be due to lower signal to noise in the hippocampus. We have amended the manuscript to include this additional discussion and analysis as follows:

Amendment to the introduction:

Furthermore, we performed an exploratory analysis to test whether effects are generalised to other brain regions (occipital, parietal and temporal lobes, dorsolateral prefrontal cortex, thalamus and amygdala) where structural and/or functional alterations have been identified in schizophrenia using *in vivo* neuroimaging³⁷⁻³⁹, and we calculated distribution volume ratio in each region of interest, using the centrum semiovale as a pseudoreference region for estimates of nondisplaceable binding of [¹¹C]UCB-J⁷.

Amendment to the method:

The CIC atlas was also used to define the dorsolateral prefrontal cortex, temporal, parietal and occipital lobes, thalamus and amygdala as additional ROIs for the exploratory analysis of effects in other brain regions. The centrum semiovale ROI was generated from the automated anatomical labelling template⁴⁰ according to parameters defined for its use as a reference region for nondisplaceable binding of [¹¹C]UCB-J⁷.

Furthermore, DVR was obtained by use of the centrum semiovale as a pseudoreference region as previously described^{7,8}.

Amendment to the results:

[¹¹C]UCB-J V_T across groups in centrum semiovale

The groups were not significantly different in [^{11}C]UCB-J V_T (ml/cm^3) in the centrum semiovale (mean [SEM] V_T (ml/cm^3) in SCZ group = 6.06 [0.34], HV group = 5.66 [0.14], $t = 1.10$, $df = 34$, $p = 0.28$).

[^{11}C]UCB-J DVR across groups in a priori ROIs

Figure 3 shows the distribution volume ratios (DVR) by group. There was a significant effect of group ($F_{1, 34} = 8.1$, $p = 0.007$) and of ROI ($F_{2, 68} = 510.9$, $p < 0.0001$) on [^{11}C]UCB-J DVR. Furthermore, there was a significant group-by-ROI interaction effect on [^{11}C]UCB-J DVR ($F_{2, 68} = 7.97$, $p = 0.0008$). Post hoc analyses with false-discovery rate (FDR) adjustment revealed that mean (SEM) [^{11}C]UCB-J DVR was significantly lower in the FC (SCZ = 2.93 [0.17]; HV = 3.48 [0.09]; $t = 2.89$, $df = 34.0$, $p = 0.01$, Cohen's $d = 1.0$), ACC (SCZ = 3.39 [0.17]; HV = 3.99 [0.09]; $t = 3.05$, $df = 34.0$, $p = 0.01$, Cohen's $d = 1.0$) and the hippocampus (SCZ = 2.40 [0.12]; HV = 2.74 [0.07]; $t = 2.32$, $df = 34.0$, $p = 0.03$, Cohen's $d = 0.8$; see Figure 3; Supplementary Figure 6).

[^{11}C]UCB-J DVR across groups in exploratory ROIs

There was a significant effect of group ($F_{1, 34} = 8.47$, $p = 0.0063$) and of ROI ($F_{5, 170} = 153.4$, $p < 0.0001$) on [^{11}C]UCB-J DVR. There was no significant group-by-ROI interaction effect on [^{11}C]UCB-J DVR ($F_{5, 170} = 1.75$, $p = 0.13$). Post hoc analyses with false-discovery rate (FDR) adjustment revealed that mean (SEM) [^{11}C]UCB-J DVR was significantly lower in the occipital lobe (SCZ = 2.93 [0.15]; HV = 3.43 [0.10]; $t = 2.72$, $df = 34.0$, $p = 0.02$, Cohen's $d = 0.9$), parietal lobe (SCZ = 2.97 [0.16]; HV = 3.46 [0.11]; $t = 2.61$, $df = 34.0$, $p = 0.02$, Cohen's $d = 0.9$), temporal lobe (SCZ = 3.10 [0.15]; HV = 3.69 [0.09]; $t = 3.36$, $df = 34.0$, $p = 0.01$,

Cohen's $d = 1.1$); dorsolateral prefrontal cortex (SCZ = 2.99 [0.17]; HV = 3.53 [0.09]; $t=2.89$, $df=34.0$, $p = 0.02$, Cohen's $d = 1.0$); thalamus (SCZ = 2.28 [0.13]; HV = 2.74 [0.10]; $t=2.77$, $df=34.0$, $p = 0.02$, Cohen's $d = 0.9$); and amygdala (SCZ = 2.95 [0.14]; HV = 3.35 [0.08]; $t=2.54$, $df=34.0$, $p = 0.02$, Cohen's $d = 0.8$, Supplementary Figures 7 and 8).

Amendment to the discussion:

Moreover, in contrast to *post-mortem* studies, we were able to examine multiple regions in each individual, and find evidence for regional differences in the magnitude of effects, given the finding of a significant group-by-region of interest interaction effect. We found lower [^{11}C]UCB-J V_T across multiple brain regions, but also evidence that frontal cortical regions may be more affected than hippocampus. The results were largely the same when using an alternative outcome measure, the distribution volume ratio (DVR), which uses a brain region, the centrum semiovale, that is thought to be largely devoid of SV2A as a reference region to adjust for non-specific binding ⁷. The one exception was the hippocampus, where DVR was significantly lower in the schizophrenia group relative to controls, albeit the effect size in the hippocampus was not as large as those in the other regions. This difference between the V_T and DVR results for the hippocampus may reflect the fact that DVR values show lower variability and so are likely to have greater sensitivity to detect group differences ^{41,42}, and our study lacked sufficient power to detect smaller effect size differences in hippocampal V_T . Thus, our finding that effects are less marked in the hippocampus relative to the frontal cortex warrants confirmation in future studies.

Reviewer #2 (Remarks to the Author):

In this interesting manuscript the authors report the first study on SV2A PET imaging in patients with schizophrenia. The study was performed by researchers which have significant experience in the field of schizophrenia and brain imaging. The experiments took a translational approach, including an evaluation of potential anti-psychotic treatment effects on SV2A examined in animals. The sample size in the clinical study (16 schizophrenia and 16 control volunteers) and the animal study (n=3-12 per group) were of reasonable size for these type of study designs. The imaging and biochemical studies were performed with appropriate methodologies and were reported in good detail. Overall, the study was of high quality and provides important insight in in vivo SV2A changes in schizophrenia (although no correlations were found with clinical symptoms).

My main comments:

1) A more detailed summary of the studied subjects (e.g. clinical assessment details) should be added.

>> We thank the reviewer for this comment. We have provided additional details of the recruitment, and clinical assessment of the patients as follows:

Methods

We recruited 18 individuals with a DSM-5 diagnosis of schizophrenia from community mental health services in London. 18 healthy volunteers were recruited through public advertisement.

and

The healthy volunteers were screened to exclude any family history of psychosis.

We have also added a new table to provide further clinical details (see supplementary table 2). Please see also the additional clinical information provided in response to reviewer 1, comment. 2.

2)there was no group effect on the corrected grey matter volumes (cGMV), which is different than several reports in the literature. Could this be related to the particular group of schizophrenia volunteers studies, could they be due to methodological differences (e.g. use of FreeSurfer) or are they related to something else?

>>Thank you for highlighting this issue. In absolute terms grey matter volumes were lower in our schizophrenia sample relative to controls but this was not significant. Meta-analyses typically report grey matter volume reductions of small-to-medium effect size and heterogeneity in the grey matter volume changes between patients^{37,43}. Studies using Freesurfer have reported significantly lower grey matter volumes in schizophrenia⁴⁴, suggesting that this method is able to detect differences. Thus, the lack of significant differences in our sample most likely reflects the relatively small changes in grey matter volume generally seen in schizophrenia coupled with heterogeneity between patients. We have now amended the discussion as follows:

Lower grey matter volumes are reported in the FC and ACC in schizophrenia³⁷, indicating partial volume effects could contribute to our findings. That we found no significant alterations in regional corrected grey matter volume likely reflects the small-to-medium effect size reductions in volume relative to healthy controls and heterogeneity in the alterations between patients³⁷, and crucially limits confounding partial volume effects.

3) V_t was used as the main outcome measure in the PET studies. It has however been shown that the white matter contains minimal SV2A, and that the centrum semiovale may be used as a reference region for [11C]UCB-J (Koole et al, 2019, EJNMI). Since V_t typically has larger inter-subject variability when compared to DVR-1, did the authors consider using a reference region approach for quantification of [11C]UCB-J? How did the V_t values of the white matter compare? It would be helpful for the reader if these values were included.

>> We thank the reviewer for this helpful comment. We have now analysed the signal as distribution volume ratio (DVR), with the centrum semiovale as a reference region and reported V_T values in this region, as suggested. Importantly these additional analyses confirm our main findings in the frontal cortex and anterior cingulate cortex. In addition, they indicate that there may be lower SV2A in hippocampus as well. Please see response to reviewer 1 point 6.

To further answer this point, regarding particularly the suitability of the centrum semiovale as a reference region for [¹¹C]UCB-J binding, we have added the following to the discussion:

Approximately 80% of [^{11}C]UCB-J V_T in nonhuman primate grey matter regions is accounted for by specific binding^{9,45}, suggesting the lower values in schizophrenia are likely to be predominantly accounted for by differences in specific binding. To investigate if non-specific uptake was influencing our findings, we repeated the analysis using DVR as the outcome, which uses the centrum semiovale as a pseudoreference region to adjust for non-specific uptake in brain. The centrum semiovale is a white matter region that is largely devoid of SV2A and displays very low uptake of [^{11}C]UCB-J^{7,8,46}. Nevertheless, there is a small amount of displacement of [^{11}C]UCB-J uptake in the centrum semiovale by a drug, levetiracetam, that is selective for SV2A⁷, suggesting that there is a degree of specific binding in the centrum semiovale. Blocking studies indicate that this is about 8% of the specific binding in grey matter, leading to a slight underestimation of specific binding in grey matter, and white matter may not be an optimal reference region because its tissue composition is different to grey matter⁴⁷. Notwithstanding this, our DVR results indicate that the lower [^{11}C]UCB-J V_T values in schizophrenia likely reflect lower specific binding to SV2A.

4) SV2A plays a role in exocytosis, and the effect of exocytosis on the binding of [^{11}C]UCB-J seems so far unclear. In the current study design, the animal study does not control for a D2R drug treatment effect on exocytosis and potential following effect on in vivo [^{11}C]UCB-J binding. The effect of exocytosis on [^{11}C]UCB-J binding should thus also be discussed as another possible reason for the group difference in V_t .

>> We thank the reviewer for this constructive feedback. We have amended our discussion of the results to include evidence that SV2A may regulate Ca^{2+} -dependent exocytosis in the presynaptic nerve terminal⁴⁸ and that the effect of exocytosis on [^{11}C]UCB-J binding is

unclear. We also acknowledge the evidence that antipsychotic drugs accumulate in synaptic vesicles and are secreted from these structures upon exocytosis, resulting in increased extracellular drug concentrations during neuronal activity, which in turn, inhibits electrically stimulated synaptic vesicle exocytosis in a dose-dependent manner⁴⁹. We note from this study that at least 14 days of haloperidol treatment, using an identical drug delivery system to our own work appears to be necessary for elevated extracellular haloperidol levels following high K⁺ stimulation⁴⁹. In the current study, we exposed rats to an identical haloperidol dose (0.5 mg/kg/d), as well as a higher dose of 2 mg/kg/d, for double the amount of time (28 days), yet we did not find any change in SV2A protein or specific binding of [³H]-UCB-J. Therefore, whilst additional experiments are required to definitively rule out any effect of antipsychotic-induced exocytosis on SV2A protein or ligand binding, the aforementioned findings suggest this is unlikely to be the case. We have amended the discussion to highlight these points as follows:

Antipsychotic drugs accumulate in synaptic vesicles and are secreted upon exocytosis, resulting in increased extracellular drug concentrations during neuronal activity and dose-dependent inhibition of electrically stimulated synaptic vesicle exocytosis⁴⁹, which could theoretically affect [¹¹C]UCB-J binding. At least 14 days of haloperidol treatment, using an identical drug delivery system to our own work, appears to be necessary for elevated extracellular haloperidol levels following high potassium stimulation⁴⁹. In the current study, we exposed rats to an identical haloperidol dose as used in this study, as well as a higher dose of 2 mg/kg/d, for double that time, but did not find any change in SV2A protein or specific binding of [³H]UCB-J. Therefore, whilst additional experiments are required to definitively rule out any effect of antipsychotic-induced vesicle exocytosis on SV2A protein or ligand binding, the aforementioned findings suggest this is unlikely to be the case.

Additional comments:

1) How were the secondary ROIs selected?

>> We have amended the method to make this clear as follows:

Furthermore, we performed an exploratory analysis to test whether effects are generalised to other brain regions (occipital, parietal and temporal lobes, dorsolateral prefrontal cortex, thalamus and amygdala) where structural and/or functional alterations have been identified in schizophrenia using *in vivo* neuroimaging³⁷⁻³⁹.

2) Where the groups matched for smoking status?

>> Subjects were not explicitly matched for smoking because there is no evidence that smoking affects SV2A levels or UCB-J uptake⁶. We have included information on smoking, conducted additional analyses to determine if this influenced findings and added a discussion of the implications. Please see responses to reviewer 1, point 2, for further details.

3) Figure 2, in the Western blot results the ratio of SV2A/GAPDH is relatively low, 0.05-

0.1. Did the authors also compare the uncorrected SV2A intensities?

>> We thank the reviewer for this constructive feedback. We have compared the uncorrected intensity of SV2A protein in the frontal cortex of the vehicle- and haloperidol-exposed rats.

These data were normally distributed (Shapiro-Wilk test $W=0.90$; $p>0.05$). One-way ANOVA revealed no statistically significant differences between the groups ($F_{2,29}=0.37$; $p=0.70$), which is in agreement with our analysis of the normalised data ($p=0.71$). We have added these to the Supplementary Notes as follows:

Supplementary Notes:

Comparison of uncorrected SV2A protein levels between vehicle- and haloperidol-exposed rats as measured by western blots

One-way ANOVA of SV2A intensity on western blot, not corrected for GAPDH signal, revealed no statistically significant differences between the groups ($F_{2,29}=0.37$; $p=0.70$)

4) Line 296, the V_t value also includes the free radiotracer concentration, in addition to the specific and nonspecific binding.

>>We thank the reviewer for providing this important comment. We have updated the discussion to make this clear as shown in the following lines to reflect the contribution of free radiotracer concentration to V_T .

It is important to recognise that V_T includes both the concentration of radioligand specifically bound to SV2A and the non-specific uptake (that is, non-specifically bound and free radioligand concentrations).

5) Line 297, the proportion of V_t reflecting specific binding relates to the regional SV2A density. In which brain region does 80% of V_t account for specific binding?

>>This is derived from occupancy plots of grey matter regions in the nonhuman primate brain⁹.

We have amended the discussion section to include the following:

Approximately 80% of [¹¹C]UCB-J V_T in nonhuman primate grey matter regions is accounted for by specific binding^{9,45}, suggesting the lower values in schizophrenia are likely to be predominantly accounted for by differences in specific binding, but blocking studies in schizophrenia are needed to confirm this.

6) Line 353, should also refer to the loss of SV2A in MDD, reference 76.

>>We thank the reviewer for this recommendation, and have duly updated the discussion to that effect as follows:

[¹¹C]UCB-J PET has demonstrated sensitivity to synaptic reductions in temporal lobe epilepsy and Alzheimer's disease^{7,18}, and has provided evidence consistent with synaptic alterations in affective disorders⁶.

7) Line 480, what time window corresponds to the 14th frame?

>> The 14th frame is acquired 540-660 seconds into the scan. We have updated the methods to reflect this as follows:

PET images were registered to each subject's MRI image and corrected for motion using frame-to-frame rigid-body registration, with the 14th frame (acquired 540-660 seconds into the scan) as the reference frame as this shows good anatomical delineation.

8) Line 486, where the brain volumes determined using FreeSurfer?

>> Brain volumes were determined using FSL (version 5.0.10; FMRIB, Oxford, UK) functions for brain extraction and Statistical and Parametric Mapping12 (Wellcome Trust Centre for Neuroimaging, <http://www.fil.ion.ucl.ac.uk/spm>) functions for image segmentation and registration via MIAKAT version 4.3.7 (<http://www.miakat.org/MIAKAT2/index.html>). We have amended the methods to make this clear as follows:

Processing and modelling were conducted using MIAKAT software version 4.3.7 (<http://www.miakat.org/MIAKAT2/index.html>), which implements functions from MATLAB (Mathworks Inc., Natick, MA, USA), FSL (version 5.0.10; FMRIB, Oxford, UK) and Statistical and Parametric Mapping12 (SPM12, Wellcome Trust Centre for Neuroimaging, <http://www.fil.ion.ucl.ac.uk/spm>).

Each subject's MRI underwent brain extraction using FSL, and grey matter segmentation and rigid body coregistration to a standard reference space⁵⁰ using SPM12 as implemented via MIAKAT. The template brain image and associated Clinical Imaging Centre (CIC) atlas⁵¹

were then warped nonlinearly to the individual subject's MRI image where the frontal cortex, anterior cingulate cortex and hippocampus were defined as the primary regions of interest (ROIs). The CIC atlas was also used to define the dorsolateral prefrontal cortex, temporal, parietal and occipital lobes, thalamus and amygdala as additional ROIs for the exploratory analysis of effects in other brain regions. The centrum semiovale ROI was generated from the automated anatomical labelling template⁴⁰ according to parameters defined for its use as a reference region for nondisplaceable binding of [¹¹C]UCB-J⁷.

PET images were registered to each subject's MRI image and corrected for motion using frame-to-frame rigid-body registration, with the 14th frame (acquired 540-660 seconds into the scan) as the reference frame as this shows good anatomical delineation. Regional time activity curves (TACs) were generated for each ROI.

Regional TAC and arterial input function data were analysed together using the one-tissue compartment model (1TCM), which has been shown to produce reliable estimates of [¹¹C]UCB-J volumes of distribution (V_T)^{46,52}.

Grey matter masks were applied to the ROIs within MIAKAT to extract both the grey matter V_T and the grey matter volume (GMV) of the ROI. For each subject, regional corrected GMV is expressed as proportions of the subject's own total brain volume, including grey matter, white matter and cerebrospinal fluid, to account for intersubject variation in total brain volume. Furthermore, regional distribution volume ratio (DVR) was obtained by use of the centrum semiovale as a pseudoreference region^{7,8}.

9) The authors interpret SV2A binding as a marker of synaptic proteins and as a proxy marker for nerve terminals. It is not discussed how SV2A is distributed across neurons from different neurotransmitters (e.g. glutamatergic vs. Gaba-ergic), and the authors should discuss this in further detail.

>> We have amended the discussion in line with the reviewer's request to more clearly elaborate on the suitability of SV2A as a marker of nerve terminals (please see response to reviewer 1, comment 5).

Reviewer #3 (Remarks to the Author):

Onwordi and colleagues present the findings of two complementary studies supporting the synaptic hypothesis in schizophrenia. In the clinical study, the authors examined 16 patients with schizophrenia and 16 healthy controls using a novel PET ligand for synaptic vesicle glycoprotein 2A (SV2A). To control for the possible effects of antipsychotic treatment, the authors performed a parallel animal study. They investigated the effects of haloperidol and olanzapine on SV2A levels in Sprague-Dawley rats, showing no significant effects. Though this is a relatively small study, the investigators present compelling in vivo evidence for the presence of decreased levels of a synaptic protein in frontal and anterior cingulate cortices in the brains of patients with schizophrenia and show that this is likely not caused by the antipsychotic treatment, but rather the disease itself. This study addresses an important question (whether patients with schizophrenia exhibit decreased levels of synaptic proteins) using a novel imaging technique. This report is an important addition

to the literature, with the following suggestions for improvement.

1. Power analyses should be presented for the human and rat studies. This is important because, at least for the human experiment, null findings (e.g., no significant differences in binding in the hippocampus) do not provide strong evidence of no group difference. More specifically, a sample size of 16 per group is only powered to detect large effect sizes. (For example, a between-group comparison using a two-sample t test has ~80% power to detect $d=1.0$ with $\alpha=0.05$.) Correlation tests of clinical variables with $n = 16$ (Table 2) are similarly limited. This limitation should be addressed in the Discussion.

>> We thank the author for this constructive feedback.

In both the human and animal experiments, there are no prior data upon which to base a power calculation, given that these are the first experiments comparing SV2A in the living human brain in healthy volunteers and patients with schizophrenia, and exploring the effects of antipsychotic drugs on SV2A in the rat brain. As such, power calculations were not performed. We have amended the paper to discuss power issues.

Please see our response to reviewer 1 (comment 5) for how we have addressed the lack of a group effect in hippocampal V_T in the discussion of the amended manuscript. We have further amended our discussion section to state the following:

Notwithstanding this, our findings in study 1 and 2 suggest that antipsychotic medication exposure is unlikely to account for the lower [^{11}C]UCB-J binding in schizophrenia patients. Nonetheless, future studies investigating SV2A in untreated, first-episode psychosis patients would be useful to test this further, and determine if SV2A changes occur early in the course

of illness. It is also important to recognise that our study is only powered to detect a strong relationship with symptoms, so it remains possible that there is a relationship between V_T and symptoms. Longitudinal studies and larger samples are needed to test the time course of alterations and their relationship with symptoms further.

2. Related to this point, "regional specificity" (line 272) is supported by the significant group-by-ROI interaction (line 115), but these results only provide evidence that the effect size for group differences in hippocampus are not "large" (lines 272-273). Indeed, Figure 1 and the p value of 0.13 suggest that the effect size in hippocampus could be small or moderate.

>> We thank the reviewer for this important comment. We agree that our findings can not exclude differences of lesser effect size. Moreover, we have conducted further analysis which also suggest the lack of significance in the hippocampus may be due to sensitivity (see response to reviewer 1 point 6).

We have therefore clarified the points made in the discussion as follows:

Moreover, in contrast to *post-mortem* studies, we were able to examine multiple regions in each individual, and find evidence for regional specificity, given the finding of a significant group-by-region of interest interaction effect.

Furthermore, we have amended the discussion to highlight that we cannot exclude a smaller difference in hippocampal [^{11}C]UCB-J V_T (please see reply to reviewer 1, comment 6).

3. Please provide images showing the a priori and exploratory regions of interest used in the human study.

>> We have added these to the Supplementary information as Supplementary Figure 10.

4. Please show individual data points in Figure 1.

>> We have provided these as requested in Supplementary Figure 1 to display individual data points, as shown here. We leave it to the Editor's discretion as to which Figure is used for Figure 1.

5. Please show effect sizes with p values, even those for which $p > 0.05$, in Figure 1 and Supplementary Figures 1 and 2.

>> As requested, we have amended these Figures such that they now display p values and effect sizes in each ROI (please see below).

6. Please provide more clinical descriptive details. This is important for reproducibility,

since clinical features of patient samples can vary widely, and those clinical features may be associated with biological measures like synaptic density. I suggest at least the following clinical variables should be described, if available: PANSS sub-scales, other concomitant medications (besides antipsychotics), age of onset of psychosis, family history of psychosis, duration of illness, nicotine/smoking status, co-morbid DSM diagnoses (anxiety, depression, OCD, past substance use disorder, etc).

>> We have added these additional descriptive details as requested where available and have provided more information on sample recruitment to aid reproducibility. Please see response to reviewer 1 point 2 for details on how we have amended the manuscript.

Regarding co-morbid DSM diagnoses, we have clarified this as follows:

Amendment to the methods

Volunteers with schizophrenia underwent a psychiatric symptom assessment by a psychiatrist using the Positive and Negative Syndrome Scale (PANSS)⁵³ to assess symptom severity and the Structured Clinical Interview for DSM-5 to confirm the diagnosis and to assess for psychiatric co-morbidities.

Amendment to the results:

All volunteers with schizophrenia were on antipsychotic medication (mean [SEM] chlorpromazine-equivalent dose = 443.5 [89.6] mg/day, Supplementary Table 1). None of the volunteers with schizophrenia had co-morbid DSM-5 psychiatric diagnoses.

7. If previous studies have validated the specificity of the SV2A polyclonal antibody and the UCB-J ligand, please cite them.

>> We thank the reviewer for highlighting this important point. The same primary antibody was used for both the western blot data and fluorescence immunostaining (rabbit polyclonal α -SV2A; ab32942, Abcam, Cambridge, UK). The selectivity of this antibody has been previously confirmed by absence of staining on slices from SV2A KO mice⁵⁴. In addition, the specificity of this antibody is confirmed by SV2A blocking peptides⁵⁴. We apologise for this oversight and have now included the appropriate references in the Supplementary Methods as follows:

Sections were then incubated for 18h at 4°C with primary antibody (Rabbit- α -SV2A, Abcam ab32942; 1:1000) diluted in blocking solution supplemented with 0.02% sodium azide.

Specificity of the SV2A antibody in immunostaining has previously been verified using slices from SV2A KO mice, as well as SV2A blocking peptides⁵⁴. Sections were then washed in PBS (3 x 10') and incubated for 2h in secondary antibody solution.

With regard to the specificity of the [¹¹C]UCB-J ligand, the following studies have validated the specificity of binding in both humans and animals in blocking experiments using levetiracetam:

Nabulsi et al., J Nucl. Med, 2016;

Finnema et al., J Cereb. Blood Flow Metab. 2018;

Bertoglio et al., J Cereb Blood Flow Metab. 2019

Furthermore, the ³H-UCB-J ligand was validated in the following studies:

Lambeng et al., Eur J Pharmacol., 2005.

Lambeng et al., Neurosci Letters, 2006

We apologise that this was not clearer and now include the appropriate references in the methods as shown below, and discuss the specificity of UCB-J further in the discussion (see response to reviewer 1, point 6):

All participants underwent a PET scan with [¹¹C]UCB-J, a radioligand specific for SV2A^{7,9,15}.

and

Individual slides containing 10-12 adjacent tissue sections at 500 µm interval were then preincubated in assay buffer (50mM Tris Base, 140mM NaCl, 1.5mM MgCl₂, 5mM KCl, 1.5mM CaCl₂, pH 7.4) for 10 min at room temperature (RT), and subsequently incubated for

2h at RT in assay buffer containing either [³H]UCB-J, a radioligand specific for SV2A^{55,56} (12.5nM), or [³H]UCB-J in the presence of levetiracetam (Sigma, Gillingham, UK; diluted to 1 mM in DMSO) to assess total and non-specific binding, respectively.

8. Nicotine use often differs between schizophrenia and control samples. Please report nicotine/tobacco use for the two groups and, if needed, address this potential confounder in the Discussion.

>> Please see our response to reviewer 1, comment 2.

9. The groups are well matched in terms of age and gender. However, both groups are predominantly male. Were sampling procedures biased toward men?

>> We thank the reviewer for highlighting this important point. There was no explicit bias towards recruiting males in the protocol. The imbalance may reflect the fact schizophrenia has a male preponderance^{57,58}, and the requirement for females to have a negative pregnancy screen and be on effective contraception during the study, which excludes some women. We have amended the discussion to discuss this issue as follows:

In study 1, the volunteers are mostly male ($n = 15$ of 18 per group), which may limit generalisations to female patients. However, previous work has not found an effect of sex on [¹¹C]UCB-J V_T ^{6,59}.

10. Are sex differences expected in the variables measured? Are UCB-J PET data available comparing women vs men? How does this imbalance affect interpretation and generalizability? This limitation should be stated in the Abstract and addressed in the Discussion.

>>In the context of animal studies we are not aware of any sexual dimorphism regarding SV2A protein levels or ligand binding. Furthermore, previous work has not detect a significant effect of sex on [¹¹C]UCB-J V_T ^{6,59}. We have added a discussion of this issue in the discussion as above (see point 9). Furthermore, in light of this important comment, within the funding available for this project we have been able to recruit and scan four additional female volunteers ($n = 2$ per group). The results essentially remain the same. We have updated the results to reflect the inclusion of more women in the study as follows:

All references to the sample size have been amended to $n = 18$, 15 male and 3 female per group.

Results

[¹¹C]UCB-J V_T across groups in a priori ROIs

There was a significant effect of group ($F_{1,34} = 6.170$, $p=0.02$) and region of interest (ROI; $F_{2,68} = 426.0$, $p < 0.0001$) on [¹¹C]UCB-J V_T . Furthermore, there was a significant group-by-ROI interaction effect on [¹¹C]UCB-J V_T ($F_{2,68} = 7.472$, $p = 0.001$). Post hoc analyses with false-discovery rate (FDR) adjustment revealed that mean (SEM) [¹¹C]UCB-J V_T (ml/cm³)

was significantly reduced in the SCZ relative to the HV group in the FC (SCZ = 16.93 [0.80]; HV = 19.50 [0.64]; $t=2.51$, $df=34.0$, $p = 0.03$), and in the ACC (SCZ = 19.55 [0.75]; HV = 22.49 [0.72]; $t=2.83$, $df=34.0$, $p = 0.02$), with large effect sizes (Cohen's $d = 0.8$ and 0.9 respectively), as shown in Figure 1 and Supplementary Figure 1. However, there was no significant difference between groups in [^{11}C]UCB-J V_T in the hippocampus (mean [SEM] V_T SCZ = 14.09 [0.59]; HV = 15.44 [0.50]; $t=1.75$, $df=34.0$, $p = 0.09$, Cohen's $d = 0.6$).

11. Line 49 – I would recommend specifying "positive" symptoms, in contrast to negative symptoms, rather than the formulation "psychotic, cognitive, and negative."

>>We thank the reviewer for this suggestion, and have updated the manuscript appropriately.

12. Line 69-71 – The authors cite studies showing lower brain volumes and altered functional connectivity in schizophrenia as indirect evidence for altered synaptic function, but there many other potential explanations for these findings, e.g., excitotoxicity/ medication effects accounting for lower volumes, white matter changes leading to altered anatomical connectivity, leading in turn to altered functional connectivity.

>>We thank the reviewer for this constructive comment. The reviewer is entirely correct to identify alternative possible causes for structural and functional alterations. It has been proposed that excessive pruning of dendritic spines contributes to deficits in grey matter volumes in schizophrenia, given the association between grey matter volumes measured by VBM and dendritic spine density⁶⁰. Functional dysconnectivity may reflect impaired integration of neuronal systems secondary to synaptic pathology¹⁴. Structural and functional

alterations may provide indirect evidence for altered synaptic function but we acknowledge there may be other explanations for these findings. We apologise for the lack of clarity and have updated the introduction to read:

Further, albeit indirect, evidence for altered synaptic function comes from *in vivo* neuroimaging studies which show lower brain volumes^{37,43,61} and altered functional connectivity in schizophrenia relative to controls⁶²⁻⁶⁴, which may reflect altered synaptic density and/or function in schizophrenia (although there are other potential explanations for these findings).

13. Line 81 - The term "synaptic dysfunction" is used in several places in the manuscript. Since the experiments presented do not measure SV2A or brain function, I would advise using this term with care. While the data do support the "synaptic dysfunction" hypothesis (Abstract), these experiments do not strongly or directly test differences in function (line 81).

>> We thank the reviewer for this important comment, and apologise for our lack of clarity. We have amended the paper to clarify this. Please see our response to reviewer 1 (comment 4).

14. Line 83 – Frontal cortex, ACC, and hippocampus are all paired structures. Did the authors hypothesize any lateralized effects?

>>We elected to explore the frontal cortex, ACC and hippocampus as paired structures on the basis of the meta-analysed postmortem literature which demonstrated reductions in synaptophysin in schizophrenia in those regions as paired structures²¹. We did not hypothesise lateralised effects and hypothesised, therefore, only that there would be reductions in SV2A in regions analogous to the those explored in the meta-analytic findings.

15. Line 85 - The meaning of "indirectly associated" is unclear.

>>We thank the reviewer for this comment and apologise for the lack of clarity here. We have amended the text to state “inversely associated”.

16. Line 108 – The authors state that “all volunteers with schizophrenia were on dopamine D2 receptor antagonist medication”. This is technically not true, since one patient is on a partial agonist, aripiprazole. It might be better to use a more general term, such as “antipsychotics”.

>>We thank the reviewer for drawing our attention to this. In light of this, we have amended the manuscript such that the term “antipsychotic drug” is used now to refer to the medications taken by the group collectively as suggested by the reviewer.

17. Line 185 – How does the length of exposure of rats – 28 days (elsewhere in the text, the authors point out this is roughly equivalent to 2.4 human years) – compare to the length of time the human subjects have been exposed to antipsychotics?

>> There is no simple answer to making age comparisons between humans and the animal models. We and others have referenced the work of Quinn⁶⁵ to address this issue. For adult rats, Quinn suggests that 11.8 days is equivalent to 1 human year, hence a 28-day exposure could be simply interpreted as equivalent to 2.4 human years of exposure. This is very likely a gross oversimplification and differences in anatomy, physiology, and developmental biology must be taken into consideration when analysing the results of any experiment. It is notable that as only 14 days of antipsychotic treatment (using identical doses and mode of administration to our work) is necessary to induce both behavioural and synaptic alterations in naïve male rats^{49,66}. Yet, after 28 days exposure in our study (double the duration) we saw no effect on SV2A protein expression or ligand binding. Nevertheless, we acknowledge that future studies should seek to validate the effect of both acute and longer duration of exposure to antipsychotic drugs to allow a definitive conclusion on this matter.

We have amended the discussion to discuss this issue as follows:

We examined the effects of a 28-day exposure (roughly equivalent to 2.4 human years⁶⁷, although this is a very simplified model). Most of the volunteers with schizophrenia in this study had taken antipsychotic drugs longer than this.

18. Lines 421-422 – Please specify what is meant by “schizophrenia spectrum disorders”

>> As defined in DSM-5, schizophrenia spectrum disorders includes schizophrenia and other schizophreniform psychotic disorders.

As all our patient subjects met DSM-5 criteria for schizophrenia, we have amended those lines to refer to schizophrenia only.

19. Line 431 – Were volunteers with schizophrenia diagnosed with the SCID, or was it a clinical diagnosis? Who performed the assessments and administered the scales? Psychiatrists? Non-MD raters? One or multiple? If there were more than one rater, did the authors measure inter-rater reliability?

>>We apologise for the lack of clarity regarding how the diagnosis was determined for volunteers with schizophrenia, and have amended the methods section as follows:

Volunteers with schizophrenia underwent a psychiatric symptom assessment by a psychiatrist using the Positive and Negative Syndrome Scale (PANSS)⁵³ to assess symptom severity and the Structured Clinical Interview for DSM-5 to confirm the diagnosis and to assess for psychiatric co-morbidities.

20. Line 777 (Supplemental Table 1) -- 5 of the 16 patients were on clozapine, which raises questions about treatment resistance, and duration of illness. Did treatment resistant patients or patients on clozapine have different VT than the non-treatment resistant ones?

>>We thank the reviewer for this important comment.

We have performed additional analyses to determine whether treatment with clozapine affected [¹¹C]UCB-J V_T. We have amended the Supplementary Notes to include the following:

[¹¹C]UCB-J V_T in clozapine-treated and non-clozapine treated schizophrenia groups in a priori ROIs

There was a significant effect of ROI ($F_{2,32} = 126.4, p < 0.0001$), but no significant effect of clozapine-treatment ($F_{1,16} = 1.86, p=0.19$) nor clozapine treatment-by-ROI interaction ($F_{2,32} = 0.215, p = 0.81$) on [¹¹C]UCB-J V_T. Post hoc analyses with false-discovery rate (FDR) adjustment revealed that [¹¹C]UCB-J V_T was not significantly different in the non-clozapine treated relative to the clozapine treated patients in any of the main regions (non-clozapine versus clozapine treated group mean (SEM) V_T (ml/cm³) for frontal cortex =17.67 [0.91] versus 15.46 [1.50]; $t=1.33, df=16.0, p = 0.28$, for the ACC = 20.14 [0.86] versus 18.38 [1.45]; $t=1.11, df=16.0, p = 0.28$ and for hippocampus = 14.74 [0.70]; versus 12.79 [0.98]; $t=1.62, df=16.0, p = 0.28$).

and we have amended the discussion to consider this as follows:

Six patients were taking clozapine, which, as this is generally reserved for patients whose illness is antipsychotic treatment resistant⁶⁸, could indicate a third of our sample met criteria for treatment resistant schizophrenia, although this was not formally assessed. Population studies indicate that antipsychotic treatment resistance is seen in about one third of chronic and about 20% of first-episode patients^{69,70}, indicating our sample is representative of chronic schizophrenia but less representative of first-episode patients in this respect. We did not find a significant difference in V_T between clozapine treated and non-clozapine treated patients, although, as the study was not designed to test differences between these groups, it would be interesting to test this in a further study.

Below are detailed additional alterations made by the authors:

>> We have updated the authors list to include an additional author, whose name was omitted from the original manuscript in error.

>>We have added group parametric [¹¹C]UCB-J V_T images as Figure 2, to provide a visual aid for the the viewer of the areas where there are differences in V_T.

>>All references to the sample size (n = 16, 15 male and 1 female per group) have been amended (to n = 18, 15 male and 3 female per group).

- 1 Mattheisen, M. *et al.* Genetic variation at the synaptic vesicle gene SV2A is associated with schizophrenia. *Schizophrenia Research* **141**, 262-265, doi:10.1016/j.schres.2012.08.027 (2012).
- 2 Kirov, G. *et al.* De novo CNV analysis implicates specific abnormalities of postsynaptic signalling complexes in the pathogenesis of schizophrenia. *Mol Psychiatry* **17**, 142-153, doi:10.1038/mp.2011.154 (2012).
- 3 Purcell, S. M. *et al.* A polygenic burden of rare disruptive mutations in schizophrenia. *Nature* **506**, 185-190, doi:10.1038/nature12975 (2014).
- 4 Fromer, M. *et al.* De novo mutations in schizophrenia implicate synaptic networks. *Nature* **506**, 179-184, doi:10.1038/nature12929 (2014).
- 5 Sekar, A. *et al.* Schizophrenia risk from complex variation of complement component 4. *Nature* **530**, 177-183, doi:10.1038/nature16549 (2016).
- 6 Holmes, S. E. *et al.* Lower synaptic density is associated with depression severity and network alterations. *Nat Commun* **10**, 1529, doi:10.1038/s41467-019-09562-7 (2019).

- 7 Finnema, S. J. *et al.* Imaging synaptic density in the living human brain. *Sci Transl Med* **8**, 348ra396, doi:10.1126/scitranslmed.aaf6667 (2016).
- 8 Mansur, A. *et al.* Characterization of 3 PET tracers for Quantification of Mitochondrial and Synaptic function in Healthy Human Brain: (18)F-BCPP-EF, (11)C-SA-4503, (11)C-UCB-J. *J Nucl Med*, doi:10.2967/jnumed.119.228080 (2019).
- 9 Nabulsi, N. B. *et al.* Synthesis and Preclinical Evaluation of 11C-UCB-J as a PET Tracer for Imaging the Synaptic Vesicle Glycoprotein 2A in the Brain. *J Nucl Med* **57**, 777-784, doi:10.2967/jnumed.115.168179 (2016).
- 10 Chang, W. P. & Sudhof, T. C. SV2 renders primed synaptic vesicles competent for Ca²⁺-induced exocytosis. *J Neurosci* **29**, 883-897, doi:10.1523/JNEUROSCI.4521-08.2009 (2009).
- 11 Crowder, K. M. *et al.* Abnormal neurotransmission in mice lacking synaptic vesicle protein 2A (SV2A). *Proceedings of the National Academy of Sciences of the United States of America* **96**, 15268-15273, doi:DOI 10.1073/pnas.96.26.15268 (1999).
- 12 Meehan, A. L., Yang, X., Yuan, L. L. & Rothman, S. M. Levetiracetam has an activity-dependent effect on inhibitory transmission. *Epilepsia* **53**, 469-476, doi:10.1111/j.1528-1167.2011.03392.x (2012).
- 13 Friston, K. J. The disconnection hypothesis. *Schizophr Res* **30**, 115-125 (1998).
- 14 Stephan, K. E., Friston, K. J. & Frith, C. D. Dysconnection in schizophrenia: from abnormal synaptic plasticity to failures of self-monitoring. *Schizophr Bull* **35**, 509-527, doi:10.1093/schbul/sbn176 (2009).
- 15 Bertoglio, D. *et al.* Validation and noninvasive kinetic modeling of [(11)C]UCB-J PET imaging in mice. *J Cereb Blood Flow Metab*, 271678X19864081, doi:10.1177/0271678X19864081 (2019).

- 16 Lynch, B. A. *et al.* The synaptic vesicle protein SV2A is the binding site for the antiepileptic drug levetiracetam. *Proc Natl Acad Sci U S A* **101**, 9861-9866, doi:10.1073/pnas.0308208101 (2004).
- 17 Bajjalieh, S. M., Frantz, G. D., Weimann, J. M., McConnell, S. K. & Scheller, R. H. Differential expression of synaptic vesicle protein 2 (SV2) isoforms. *The Journal of Neuroscience* **14**, 5223-5235, doi:10.1523/jneurosci.14-09-05223.1994 (1994).
- 18 Chen, M. K. *et al.* Assessing Synaptic Density in Alzheimer Disease With Synaptic Vesicle Glycoprotein 2A Positron Emission Tomographic Imaging. *JAMA Neurol* **75**, 1215-1224, doi:10.1001/jamaneurol.2018.1836 (2018).
- 19 Mutch, S. A. *et al.* Protein quantification at the single vesicle level reveals that a subset of synaptic vesicle proteins are trafficked with high precision. *J Neurosci* **31**, 1461-1470, doi:10.1523/JNEUROSCI.3805-10.2011 (2011).
- 20 Mudge, J. *et al.* Genomic convergence analysis of schizophrenia: mRNA sequencing reveals altered synaptic vesicular transport in post-mortem cerebellum. *PLoS One* **3**, e3625, doi:10.1371/journal.pone.0003625 (2008).
- 21 Osimo, E. F., Beck, K., Reis Marques, T. & Howes, O. D. Synaptic loss in schizophrenia: a meta-analysis and systematic review of synaptic protein and mRNA measures. *Molecular Psychiatry*, doi:10.1038/s41380-018-0041-5 (2018).
- 22 Eastwood, S. L. & Harrison, P. J. Decreased Synaptophysin in the Medial Temporal Lobe in Schizophrenia Demonstrated Using Immunohistochemistry. *Neuroscience* **69**, 339-343, doi:10.1016/0306-4522(95)00324-C (1995).
- 23 Perrone-Bizzozero, N. I. *et al.* Levels of the growth-associated protein GAP-43 are selectively increased in association cortices in schizophrenia. *Proc Natl Acad Sci U S A* **93**, 14182-14187 (1996).

- 24 Glantz, L. A. Reduction of Synaptophysin Immunoreactivity in the Prefrontal Cortex of Subjects With Schizophrenia. *Archives of General Psychiatry* **54**, doi:10.1001/archpsyc.1997.01830220065010 (1997).
- 25 Davidsson, P. *et al.* The synaptic-vesicle-specific proteins rab3a and synaptophysin are reduced in thalamus and related cortical brain regions in schizophrenic brains. *Schizophrenia Research* **40**, 23-29, doi:Doi 10.1016/S0920-9964(99)00037-7 (1999).
- 26 Matosin, N. *et al.* Molecular evidence of synaptic pathology in the CA1 region in schizophrenia. *NPJ Schizophr* **2**, 16022, doi:10.1038/npjpsz.2016.22 (2016).
- 27 Selemon, L. D. & Goldman-Rakic, P. S. The reduced neuropil hypothesis: a circuit based model of schizophrenia. *Biol Psychiatry* **45**, 17-25 (1999).
- 28 Glantz, L. A. & Lewis, D. A. Decreased dendritic spine density on prefrontal cortical pyramidal neurons in schizophrenia. *Arch Gen Psychiatry* **57**, 65-73 (2000).
- 29 Glausier, J. R. & Lewis, D. A. Dendritic spine pathology in schizophrenia. *Neuroscience* **251**, 90-107, doi:10.1016/j.neuroscience.2012.04.044 (2013).
- 30 MacDonald, M. L. *et al.* Selective Loss of Smaller Spines in Schizophrenia. *Am J Psychiatry* **174**, 586-594, doi:10.1176/appi.ajp.2017.16070814 (2017).
- 31 Moyer, C. E., Shelton, M. A. & Sweet, R. A. Dendritic spine alterations in schizophrenia. *Neurosci Lett* **601**, 46-53, doi:10.1016/j.neulet.2014.11.042 (2015).
- 32 Harrison, P. J. & Weinberger, D. R. Schizophrenia genes, gene expression, and neuropathology: on the matter of their convergence. *Mol Psychiatry* **10**, 40-68; image 45, doi:10.1038/sj.mp.4001558 (2005).
- 33 Feinberg, I. Schizophrenia: caused by a fault in programmed synaptic elimination during adolescence? *J Psychiatr Res* **17**, 319-334 (1982).
- 34 Keshavan, M. S., Anderson, S. & Pettegrew, J. W. Is Schizophrenia due to excessive synaptic pruning in the prefrontal cortex? The Feinberg hypothesis revisited. *Journal*

of *Psychiatric Research* **28**, 239-265, doi:[https://doi.org/10.1016/0022-3956\(94\)90009-4](https://doi.org/10.1016/0022-3956(94)90009-4) (1994).

- 35 Howes, O. D. & McCutcheon, R. Inflammation and the neural diathesis-stress hypothesis of schizophrenia: a reconceptualization. *Translational Psychiatry* **7**, e1024-e1024, doi:10.1038/tp.2016.278 (2017).
- 36 Sellgren, C. M. *et al.* Increased synapse elimination by microglia in schizophrenia patient-derived models of synaptic pruning. *Nat Neurosci*, doi:10.1038/s41593-018-0334-7 (2019).
- 37 Brugger, S. P. & Howes, O. D. Heterogeneity and Homogeneity of Regional Brain Structure in Schizophrenia: A Meta-analysis. *JAMA Psychiatry* **74**, 1104-1111, doi:10.1001/jamapsychiatry.2017.2663 (2017).
- 38 O'Neill, A., Mechelli, A. & Bhattacharyya, S. Dysconnectivity of Large-Scale Functional Networks in Early Psychosis: A Meta-analysis. *Schizophr Bull* **45**, 579-590, doi:10.1093/schbul/sby094 (2019).
- 39 Rimol, L. M. *et al.* Cortical thickness and subcortical volumes in schizophrenia and bipolar disorder. *Biol Psychiatry* **68**, 41-50, doi:10.1016/j.biopsych.2010.03.036 (2010).
- 40 Tzourio-Mazoyer, N. *et al.* Automated Anatomical Labeling of Activations in SPM Using a Macroscopic Anatomical Parcellation of the MNI MRI Single-Subject Brain. *NeuroImage* **15**, 273-289, doi:10.1006/nimg.2001.0978 (2002).
- 41 Bloomfield, P. S. *et al.* Microglial Activity in People at Ultra High Risk of Psychosis and in Schizophrenia: An [(11)C]PBR28 PET Brain Imaging Study. *Am J Psychiatry* **173**, 44-52, doi:10.1176/appi.ajp.2015.14101358 (2016).
- 42 Lyoo, C. H. *et al.* Cerebellum Can Serve As a Pseudo-Reference Region in Alzheimer Disease to Detect Neuroinflammation Measured with PET Radioligand Binding to

- Translocator Protein. *J Nucl Med* **56**, 701-706, doi:10.2967/jnumed.114.146027 (2015).
- 43 Haijma, S. V. *et al.* Brain volumes in schizophrenia: a meta-analysis in over 18 000 subjects. *Schizophr Bull* **39**, 1129-1138, doi:10.1093/schbul/sbs118 (2013).
- 44 van Erp, T. G. *et al.* Subcortical brain volume abnormalities in 2028 individuals with schizophrenia and 2540 healthy controls via the ENIGMA consortium. *Mol Psychiatry* **21**, 547-553, doi:10.1038/mp.2015.63 (2016).
- 45 Rabiner, E. A. Imaging Synaptic Density: A Different Look at Neurologic Diseases. *J Nucl Med* **59**, 380-381, doi:10.2967/jnumed.117.198317 (2018).
- 46 Finnema, S. J. *et al.* Kinetic evaluation and test-retest reproducibility of [(11)C]UCB-J, a novel radioligand for positron emission tomography imaging of synaptic vesicle glycoprotein 2A in humans. *J Cereb Blood Flow Metab* **38**, 2041-2052, doi:10.1177/0271678X17724947 (2018).
- 47 Rossano, S. *et al.* Assessment of a white matter reference region for (11)C-UCB-J PET quantification. *J Cereb Blood Flow Metab*, 271678X19879230, doi:10.1177/0271678X19879230 (2019).
- 48 Xu, T. & Bajjalieh, S. M. SV2 modulates the size of the readily releasable pool of secretory vesicles. *Nat Cell Biol* **3**, 691-698, doi:10.1038/35087000 (2001).
- 49 Tischbirek, C. H. *et al.* Use-dependent inhibition of synaptic transmission by the secretion of intravesicularly accumulated antipsychotic drugs. *Neuron* **74**, 830-844, doi:10.1016/j.neuron.2012.04.019 (2012).
- 50 Grabner, G. *et al.* in *Medical Image Computing and Computer-Assisted Intervention – MICCAI 2006*. (eds Rasmus Larsen, Mads Nielsen, & Jon Sporring) 58-66 (Springer Berlin Heidelberg).

- 51 Tziortzi, A. C. *et al.* Imaging dopamine receptors in humans with [11C]-(+)-PHNO: dissection of D3 signal and anatomy. *Neuroimage* **54**, 264-277, doi:10.1016/j.neuroimage.2010.06.044 (2011).
- 52 Mansur A, R. E., Comley RA, Lewis Y, Middleton LT, Huiban M, Passchier J, Tsukada H, Gunn RN and MIND MAPS Consortium. Characterization of 3 PET tracers for Quantification of Mitochondrial and Synaptic function in Healthy Human Brain: [18F]BCPP-EF, [11C]SA-4503, [11C]UCB-J. *In press* (2019).
- 53 Kay, S. R., Fiszbein, A. & Opler, L. A. The Positive and Negative Syndrome Scale (PANSS) for Schizophrenia. *Schizophrenia Bulletin* **13**, 261-276, doi:10.1093/schbul/13.2.261 (1987).
- 54 Crevecoeur, J. *et al.* Expression of SV2 isoforms during rodent brain development. *BMC Neurosci* **14**, 87, doi:10.1186/1471-2202-14-87 (2013).
- 55 Lambeng, N., Gillard, M., Vertongen, P., Fuks, B. & Chatelain, P. Characterization of [(3)H]ucb 30889 binding to synaptic vesicle protein 2A in the rat spinal cord. *Eur J Pharmacol* **520**, 70-76, doi:10.1016/j.ejphar.2005.07.029 (2005).
- 56 Lambeng, N., Grossmann, M., Chatelain, P. & Fuks, B. Solubilization and immunopurification of rat brain synaptic vesicle protein 2A with maintained binding properties. *Neurosci Lett* **398**, 107-112, doi:10.1016/j.neulet.2005.12.059 (2006).
- 57 Aleman, A., Kahn, R. S. & Selten, J. P. Sex differences in the risk of schizophrenia: evidence from meta-analysis. *Arch Gen Psychiatry* **60**, 565-571, doi:10.1001/archpsyc.60.6.565 (2003).
- 58 McGrath, J., Saha, S., Chant, D. & Welham, J. Schizophrenia: a concise overview of incidence, prevalence, and mortality. *Epidemiol Rev* **30**, 67-76, doi:10.1093/epirev/mxn001 (2008).

- 59 Carson, R. *et al.* Age and sex effects on synaptic density in healthy humans as assessed with SV2A PET. *Journal of Nuclear Medicine* **59**, 541 (2018).
- 60 Keifer, O. P., Jr. *et al.* Voxel-based morphometry predicts shifts in dendritic spine density and morphology with auditory fear conditioning. *Nat Commun* **6**, 7582, doi:10.1038/ncomms8582 (2015).
- 61 Steen, R. G., Mull, C., McClure, R., Hamer, R. M. & Lieberman, J. A. Brain volume in first-episode schizophrenia: systematic review and meta-analysis of magnetic resonance imaging studies. *Br J Psychiatry* **188**, 510-518, doi:10.1192/bjp.188.6.510 (2006).
- 62 Lynall, M. E. *et al.* Functional connectivity and brain networks in schizophrenia. *J Neurosci* **30**, 9477-9487, doi:10.1523/JNEUROSCI.0333-10.2010 (2010).
- 63 Fornito, A., Yoon, J., Zalesky, A., Bullmore, E. T. & Carter, C. S. General and specific functional connectivity disturbances in first-episode schizophrenia during cognitive control performance. *Biol Psychiatry* **70**, 64-72, doi:10.1016/j.biopsych.2011.02.019 (2011).
- 64 Woodward, N. D., Rogers, B. & Heckers, S. Functional resting-state networks are differentially affected in schizophrenia. *Schizophr Res* **130**, 86-93, doi:10.1016/j.schres.2011.03.010 (2011).
- 65 Quinn, R. Comparing rat's to human's age: how old is my rat in people years?
- 66 Amato, D. *et al.* A dopaminergic mechanism of antipsychotic drug efficacy, failure, and failure reversal: the role of the dopamine transporter. *Mol Psychiatry*, doi:10.1038/s41380-018-0114-5 (2018).
- 67 Quinn, R. Comparing rat's to human's age: how old is my rat in people years? *Nutrition* **21**, 775-777, doi:10.1016/j.nut.2005.04.002 (2005).

- 68 Mouchlianitis, E., McCutcheon, R. & Howes, O. D. Brain-imaging studies of treatment-resistant schizophrenia: a systematic review. *The Lancet Psychiatry* **3**, 451-463, doi:10.1016/s2215-0366(15)00540-4 (2016).
- 69 Lally, J. *et al.* Two distinct patterns of treatment resistance: clinical predictors of treatment resistance in first-episode schizophrenia spectrum psychoses. *Psychol Med* **46**, 3231-3240, doi:10.1017/S0033291716002014 (2016).
- 70 Demjaha, A. *et al.* Antipsychotic treatment resistance in first-episode psychosis: prevalence, subtypes and predictors. *Psychol Med* **47**, 1981-1989, doi:10.1017/S0033291717000435 (2017).

REVIEWERS' COMMENTS:

Reviewer #1 (Remarks to the Author):

The authors have addressed my comments satisfactorily.

Reviewer #2 (Remarks to the Author):

The authors have satisfactorily addressed my concerns.

Reviewer #3 (Remarks to the Author):

The authors have thoroughly addressed all of my criticisms.